# High-speed mapping of surface charge dynamics using sparse scanning Kelvin probe force microscopy

Marti Checa [1] ✉, Addis S. Fuhr[1], Changhyo Sun[2], Rama Vasudevan [1], Maxim Ziatdinov [1,3], Ilia Ivanov [1], Seok Joon Yun [1,7], Kai Xiao [1], Alp Sehirlioglu [4], Yunseok Kim [2], Pankaj Sharma [5,6], Kyle P. Kelley [1], Neus Domingo[1], Stephen Jesse[1] & Liam Collins[1] ✉

Unraveling local dynamic charge processes is vital for progress in diverse fields, from microelectronics to energy storage. This relies on the ability to map charge carrier motion across multiple length- and timescales and understanding how these processes interact with the inherent material heterogeneities. Towards addressing this challenge, we introduce high-speed sparse scanning Kelvin probe force microscopy, which combines sparse scanning and image reconstruction. This approach is shown to enable sub-second imaging (>3 frames per second) of nanoscale charge dynamics, representing several orders of magnitude improvement over traditional Kelvin probe force microscopy imaging rates. Bridging this improved spatiotemporal resolution with macroscale device measurements, we successfully visualize electrochemically mediated diffusion of mobile surface ions on a $LaAlO_3$/ $SrTiO_3$ planar device. Such processes are known to impact band-alignment and charge-transfer dynamics at these heterointerfaces. Furthermore, we monitor the diffusion of oxygen vacancies at the single grain level in polycrystalline $TiO_2$. Through temperature-dependent measurements, we identify a charge diffusion activation energy of 0.18 eV, in good agreement with previously reported values and confirmed by DFT calculations. Together, these findings highlight the effectiveness and versatility of our method in understanding ionic charge carrier motion in microelectronics or nanoscale material systems.

The investigation of charge dynamics in nanoscale systems holds paramount importance for advancing next-generation materials and devices. Of major significance is understanding the motion and transport of ionic charge carriers in energy technologies like batteries and fuel cells, as well as computing and memory devices, such as memristors and ferroelectrics, to name but a few. While the pervasive nature of ionic processes in modern technology underscores their significance, the growing trend of miniaturization and utilization of low dimensional materials presents additional challenges for existing characterization techniques[1,2].

[1]Center for Nanophase Materials Sciences, Oak Ridge National Laboratory, Oak Ridge, TN 37831, USA. [2]School of Advanced Materials Science and Engineering, Sungkyunkwan University, Suwon 16419, Republic of Korea. [3]Computational Sciences and Engineering Division, Oak Ridge National Laboratory, Oak Ridge, TN 37923, USA. [4]Department of Materials Science and Engineering, Case Western Reserve University, Cleveland, OH 44106, USA. [5]College of Science and Engineering, Flinders University, Bedford Park, SA 5042, Australia. [6]ARC Centre of Excellence in Future Low-Energy Electronics Technologies (FLEET), UNSW Sydney, Sydney, NSW 2052, Australia. [7]Present address: Department of Semiconductor, University of Ulsan, Ulsan 44610, Korea. ✉e-mail: checam@ornl.gov; collinslf@ornl.gov

Electrostatic force microscopy (EFM) and Kelvin probe force microscopy (KPFM) are widely recognized as the gold standards for nanoscale imaging of surface potentials across various materials, including electronic junctions[3], ionic conductors[4], biological samples[5,6], and opto-electronic devices[7–9] amongst a myriad of other applications. These techniques are known to provide valuable insights into surface charges, dielectric properties, and the contact potential difference (CPD), which represents the difference in work function between the tip and the sample. Pushing the spatial resolution of these techniques, has allowed local potential characterization to be performed at the atomic level[10,11], enabling imaging charge distribution within a single molecule[12,13]. Indeed, traditional EFM/KPFM techniques offer valuable insights into the static relationship between structure and properties in nanoscale systems. However, their ability to capture dynamics is limited due to the inherently slow scanning speed of scanning probe microscopies (SPM). As a result, crucial dynamic phenomena like charge screening, migration, and diffusion of ionic charge carriers, as well as fast electrochemical processes remain inaccessible, hindering a comprehensive understanding of many important device mechanisms and processes. To address this limitation, there is a need for innovative and faster scanning methods that can enable real-time observations of dynamic processes across appropriate length scales[14].

In this study, we introduce sparse scanning KPFM (SS-KPFM), which combines sparse scanning trajectories with advanced image reconstruction using Gaussian processing (GP) in-painting, resulting in significantly improved imaging speeds. We demonstrate that SS-KPFM can recover the CPD with good accuracy while maintaining high spatial resolution. SS-KPFM sets a modern standard in KPFM imaging rates, surpassing traditional KPFM imaging speeds by over 3 orders of magnitude. This advancement unlocks the potential to probe nanoscale charge dynamics previously hidden due to limited imaging bandwidth, offering deeper insights into material behaviors and device operations. Utilizing SS-KPFM, we demonstrate the possibility to successfully bridge nanoscale surface charge dynamics with macroscale device characterization on a technologically relevant LaAlO$_3$/SrTiO$_3$ (LAO/STO) device featuring planar electrodes. Our findings reveal the substantial contribution of electrochemically mediated diffusion of mobile surface ions on timescales of tens of seconds. To further validate the approach, we demonstrate its capability to monitor ionic charge carriers, with our nanoscopic probe functioning both as a nano-electrode and sensor. This especial capability is demonstrated through the study of temperature-dependent oxygen vacancy motion in a polycrystalline TiO$_2$ sample at the single grain level. The results reveal Arrhenius-like processes, characterized by an activation energy of 0.18 eV, which are in good agreement with reported values of oxygen vacancy diffusion and further supported by additional molecular dynamics (MD) simulations. These findings underscore the utility and potential of SS-KPFM for investigating nanoscale ion dynamics and electrochemical processes in various technologically relevant systems.

## Results and discussion
### Towards high speed KPFM
First it is important to briefly review the recent advancement of time-resolved EFM (tr-EFM) and time-resolved KPFM (tr-KPFM) methods which enable the study of certain charge dynamics in materials at the nanoscale[15]. Here, we make an important distinction between time-resolved and high-speed imaging, acknowledging that both capture spatiotemporal dynamics. Specifically, time resolved methods typically involve slowly scanning, or placing, a tip above a specific location of interest while rapidly perturbing the sample or device at regular time intervals using an external stimulus (e.g., often bias, or light). In time-resolved experiments, the sample often undergoes multiple global perturbations while the probe measures the local response at a specific nanoscale location before moving on to the next location. To date, tr-

EFM and tr-KPFM have been applied to a variety of systems, including optoelectronic materials[16–18], semiconductors for transistors[19–21] or memristors[22], with temporal resolution ranging from 10 s to 1 ps[23–27]. While these approaches provide a detailed record of how the sample's properties evolve at each location over time, they are typically limited to fast processes that are fully reversible within the measurement window. In other words, these methods are limited to the study of fully reversible processes that quickly equilibrate within the measurement window (cyclo-stationary).

To capture spatiotemporal dynamics of slower (≈100 s to ms) or non-cyclo-stationary processes we propose the implementation of high-speed KPFM. Unlike time resolved approaches, high speed imaging facilitates the rapid acquisition of multiple images/frames per second without the need for repeated perturbation/excitation of the system under test. This facilitates a comprehensive picture of how properties evolve in both space and time by analyzing differences among rapidly acquired images. This schema is particularly beneficial for the investigation of samples that cannot be excited at regular intervals or in the study of non-cyclo-stationary, non-ergodic[28], or irreversible events. For example, high-speed KPFM could shed light on irreversible transformations, such as fatigue in ferroelectric capacitors[29], the nanoscale origins of corrosion in metals and alloys[30,31], the formation of a solid-electrolyte interface layer in a battery[32] or dendrite filament formation in memristors[33,34], to name but a few.

Recently, Garret et al.[35] have presented an innovative advancement in the field of KPFM. By implementing a frequency modulated (FM) variant of heterodyne-KPFM (H-KPFM) they demonstrated fast scanning at a rate of 16 s per frame, an improvement in imaging speed of more than 100-fold compared to conventional FM-KPFM. This set a benchmark for high-speed KPFM. For the sake of readers, we have included a Supplementary Fig. 1 containing a comprehensive summary of the development and operational principles of all heterodyne KPFM variants. In the FM variant of KPFM (which we refer to as H-KPFM), the electrical excitation ($\omega_E$) is performed at $\omega_E = \omega_1 - \omega_0$, which shifts the sideband frequency to the second eigenmode at $\omega_1$, where $\omega_0$ and $\omega_1$ are the frequencies of the first and second cantilever eigenmodes, respectively. This methodology has significant advantages including high sensitivity to the electrostatic force gradient (eliminating stray capacitance effect[36]), as well as higher detection bandwidth than other implementations of KPFM[35]. The resolution enhancement offered by H-KPFM compared to classical amplitude modulated KPFM, is demonstrated in Fig. S1 for a KPFM calibration sample based on interdigitated Al and Au line electrodes deposited on glass (Budget Sensors KPFM & EFM calibration sample). In a separate study, Garret et al.[37] employed this fast-scanning H-KPFM approach to spatially map the real-time open circuit voltage dynamics of hybrid organic-inorganic perovskites based on methylammonium lead iodide (MAPbI$_3$), highlighting the potential for this technique to greatly enhance the spatial and temporal resolution of nanoscale functional imaging. However, the authors noted that surpassing the 16 seconds per frame (0.0625 fps) benchmark was impeded by the limitation of the topographic feedback loop, a common limitation in most functional SPM or KPFM measurements[35].

To address the speed constraints in imaging due to feedback restrictions and uneven accelerations in traditional raster scans, researchers have explored the use of non-rectangular scanning methods. Several studies[38–42] have investigated non-rectangular scans, aiming to overcome topographic feedback challenges and ultimately enhance the overall imaging speed. In most of these implementations, tip trajectory waveforms are directly applied to the voltage input of the XY piezo scanners, operating them in an open-loop configuration. By utilizing desired non-rectangular scanning paths, such as spiral scanning[41] or Lissajous scan paths[43], researchers have demonstrated the possibility to mitigate the feedback limitations and reduce the non-uniformity of accelerations experienced by the moving probe during

imaging. These strategies have significantly increased the achievable imaging speed in SPM, as demonstrated by examples of tracking atomic diffusion by scanning tunneling microscopy (STM)[40] and imaging electrochemical fluxes by scanning electrochemical cell microscopy (SECCM)[44], among other applications[45,46,54].

In entirely separate fields of research, the principle of sparse sampling has been critical to characterization tools in diverse fields such medical imaging[47], electron microscopy[48] as well as astronomy[49]. For clarity purposes, we refer to the term *sparse* in its mathematical definition; that is an image (matrix) in which most measurement locations are zero (i.e., unscanned regions). By exploiting the sparsity of the image, researchers can enhance data acquisition efficiency by collecting only a fraction of the total measurements. Subsequently, specialized algorithms are employed to reconstruct the entire image from these sparse measurements, effectively filling in the unscanned regions. Fortunately, great strides are being made in the advent of machine learning image reconstruction algorithms such as compressed sensing (CS)[38], convolutional neural networks[50] (CNN) or Gaussian process (GP) optimization[51,52], which are critical to the accurate reconstruction of images captured by sparse data schemes. Although not widely adopted to date, combining sparse, non-rectangular scans, with image reconstruction algorithms offer several advantages to SPM, including minimized perturbation to the system,

gentler acceleration paths for the piezo positioners that move the tip (avoiding aggressive accelerations at the end of each line), extended probe life, and importantly their combination presents clear opportunities for faster image acquisition. Indeed, this strategy has recently been successfully implemented by our team to enhance the scanning speed of piezoresponse force microscopy (PFM)[53,54], but no such implementation has been realized for KPFM.

## Sparse scanning KPFM

In Fig. 1a, we present the schematics of the SS-KPFM experimental setup. Briefly, the integration of a commercial AFM system with a custom-built field programmable gate array (FPGA), we achieve precise control over the piezo motion. This allows us to produce custom voltage waveforms. As a result, we can direct the piezo positioners to follow specific motion trajectories, like a spiral scan. A more detailed explanation of the experimental setup is shown in the Supplementary Fig. 2 and in the materials section. Notably, while this work primarily focuses on the use of sparse spiral scans, our method allows for the execution of any desired scan path by generating the corresponding *XY* piezo voltage trajectories. An example of a Lissajous curve scan path can be found in the Supplementary Fig. 3.

In Fig. 1b, the raw data from an SS-KPFM image across a WS2 monolayer flake on a highly p-doped silicon substrate in shown.

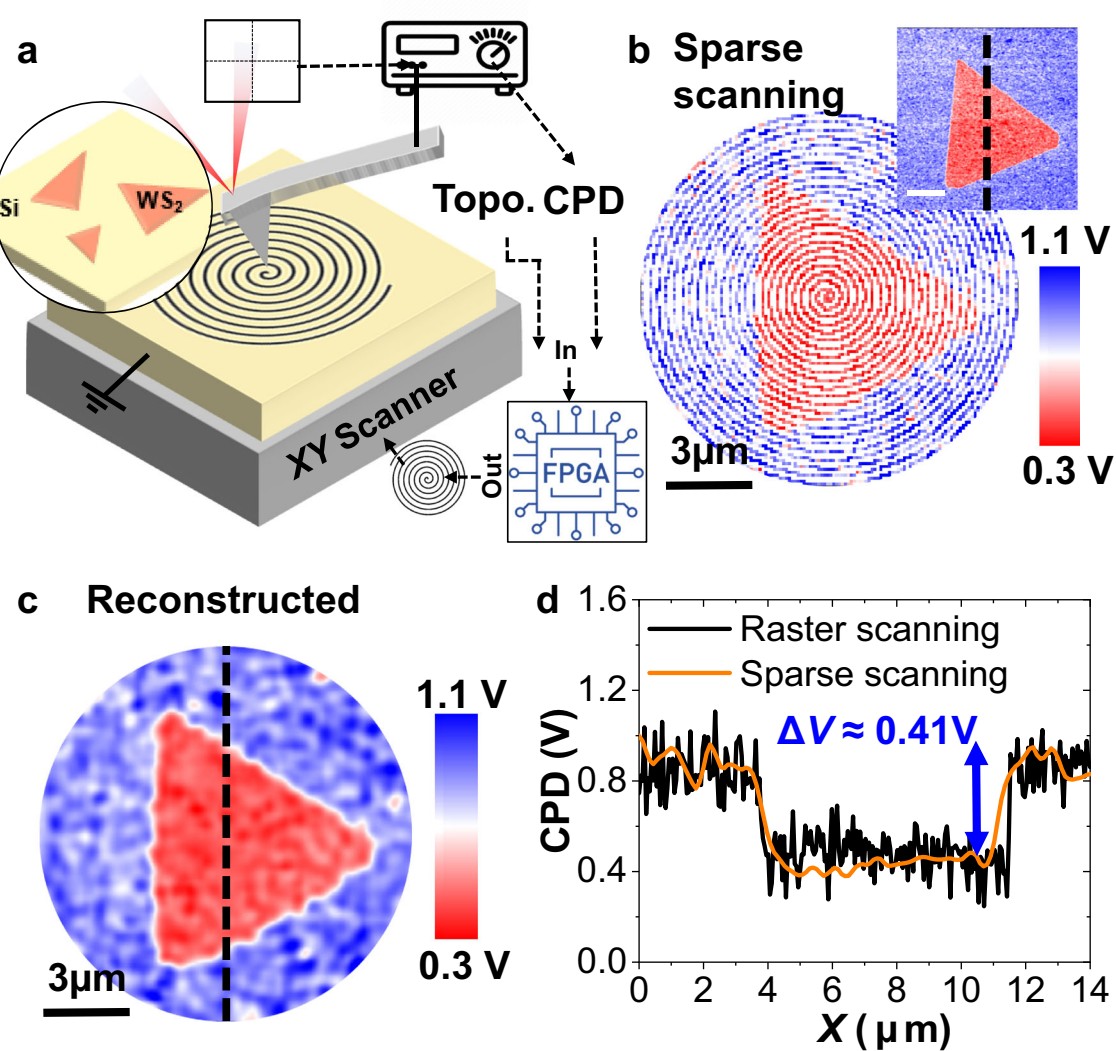

**Fig. 1 | Experimental setup and technique validation. a** Experimental setup showing the tip trajectory during one SS-KPFM scan. **b** SS-KPFM Raw data of a WS2 flake grown on a p-doped Si substrate at 0.25 fps. Inset shows raster scan H-KPFM acquired in the same flake at 0.000065 fps. Scalebar for inset is also 3 μm. **c** Fully reconstructed image using GP. **d** Comparison of the H-KPFM profiles using raster scan H-KPFM and SS-KPFM.

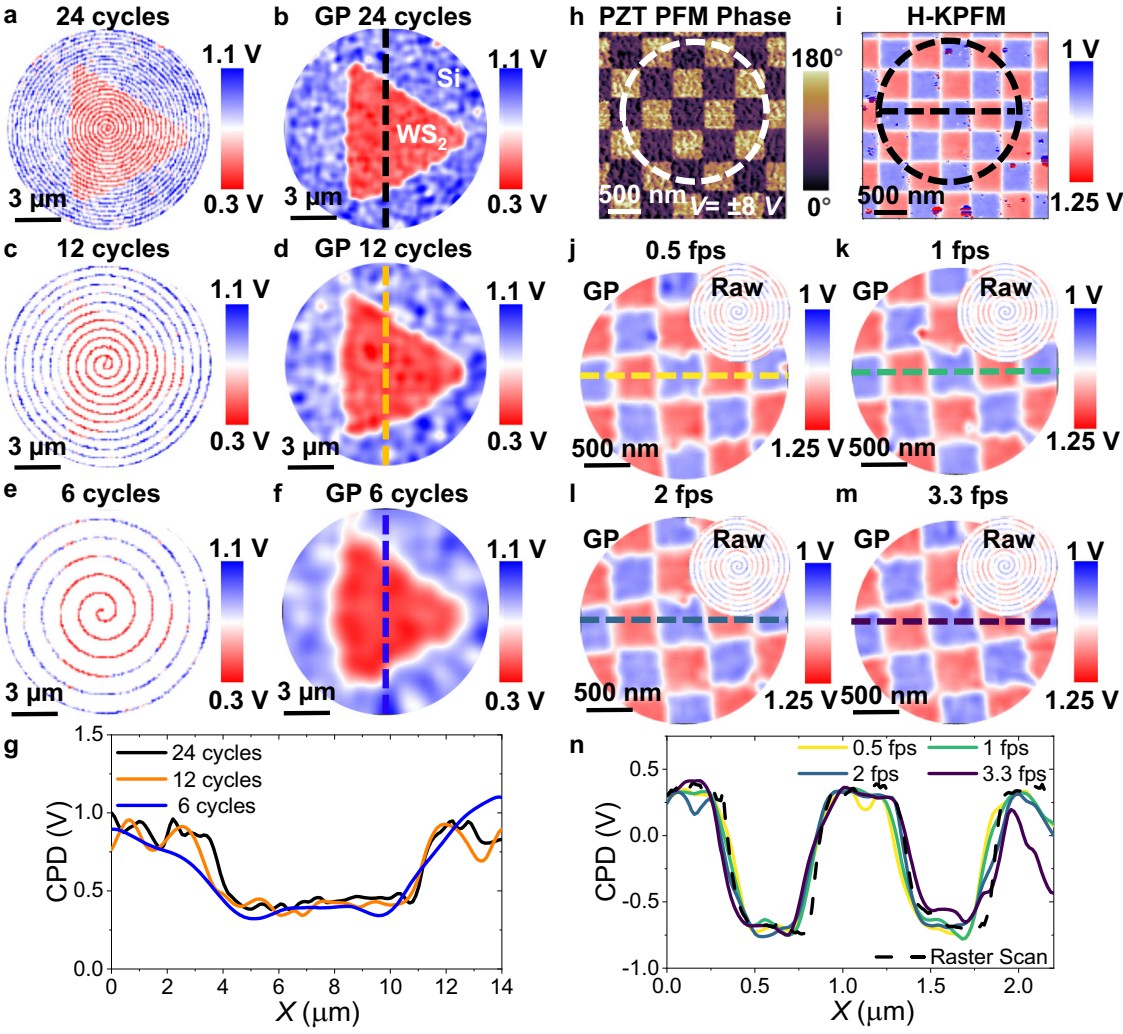

**Fig. 2 | Image reconstruction in SS-KPFM: Sparsity and scanning speed.** SS-KPFM images of the 2D WS$_2$ triangular flake on Si substrate for different number of spiral cycles, **a** 24 cycles, **c** 12 cycles, **e** 6 cycles. Corresponding GP reconstructed images in (**b**), (**d**), and (**f**). **g** CPD profile along the dashed lines (black 24 cycles, orange 12 cycles and blue 6 cycles). Images of the checkerboard written in a PZT thin film: **h** Single Frequency PFM Phase Image. **i** Raster scan H-KPFM at 0.000065 fps. SS-KPFM at: **j** 0.5 fps, **k** 1 fps, **l** 2 fps, **m** 3.3 fps. Insets are the SS-KPFM Raw data for each case. **n** Corresponding CPD profiles along the dashed lines (dashed black raster scan, yellow 0.5 fps, green 1 fps, turquoise blue 2 fps, dark blue 3.3 fps).

The image was acquired using a spiral trajectory with 24 internal cycles and an imaging speed of 4 s per frame (0.25 fps). The inset shows the same region scanned using single pass H-KPFM using traditional raster scanning, which took 4 min and 16 s to collect (0.000065 fps). Figure 1c displays the reconstructed CPD image, where the missing pixels have been filled in using GP for image reconstruction. GP estimates an unknown non-linear function from noisy observations of that function at a finite number of points, assuming that the observations are a sample from the multivariate Gaussian distribution. This approach enables us to efficiently reconstruct the missing pixels and obtain a complete CPD map (explained in further detail in ref. 53). The observations are linked via the kernel whose parameters can be learned during the regression process. In Fig. 1d, we quantitatively compare the CPD measured using the fast SS-KPFM and classic H-KPFM, validating the method by measuring a consistent value of $\Delta V \approx 0.41$ V between the silicon substrate and the WS$_2$ flake. Although the signal-to-noise ratio (SNR) appears better for SS-KPFM than raster H-KPFM in Fig. 1d, it is important to note that the GP itself acts as a filter, resulting in an averaging of the measured signal in the neighboring pixels, which has the effect of reducing the noise. A more detailed analysis of the SNR can be found in Supplementary Fig. 4, where we compare the GP reconstruction performance on SS-KPFM data versus masked raster scan.

Next, we conduct a comprehensive exploration of the spiral scan and image reconstruction parameters, analyzing their influence on imaging rate and spatial resolution. In Fig. 2, we present SS-KPFM images of the same WS$_2$ flake as shown in Fig. 1, while varying the number of spiral cycles from 24 to 6. The GP image reconstruction algorithm uses the "Matern52" kernel function and a learning rate of 0.2[53]. The correlation length parameter is adjusted based on the sparsity of scans; sparser scans are reconstructed using a longer correlation length. Further information on these parameters is available in Supplementary Figs. 4 and 5. The sequence of figures shown in Fig. 2a–f clearly illustrates the strengths and limitations of the GP reconstruction algorithm when dealing with sparse scans. As the sparsity (number of cycles) in the measured scan increased (decreases), there is less data available for the image reconstruction algorithm, resulting in a less accurate reconstruction. Even with only 6 cycles of the spiral scan, the final image successfully recovers the triangular shape of the WS$_2$ flake. However, it is evident that the image appears significantly blurrier compared to higher density scans. Moreover, as shown in Fig. 2g, the recovered CPD is quantitatively accurate in all cases, although the fine spatial resolution decreases with increasing sparsity of measurements. Finally, an important aspect to consider is that the sparsity required for dependable image reconstruction is

influenced by the form and dimensions of the structures being imaged. As a result, sparsity can be adjusted based on the sample type, optimizing the balance between scanning speed and spatial resolution. All in all, these results highlight the effectiveness and robustness of the SS-KPFM technique in capturing essential features with a significantly reduced scanning duration, offering potential benefits for faster imaging and analysis of nanoscale spatiotemporal dynamics.

To evaluate the ultimate imaging speed of the SS-KPFM technique, we selected a model sample consisting of written ferroelectric domains with opposite polarization on a lead zirconate titanate thin film (PZT). Using a writing bias of ±8 V, we patterned a checkerboard-like design to create an ideal test sample with localized charge regions and no associated topographic information. After patterning the ferroelectric domains, we conducted a PFM image acquired in normal raster scan mode to identify the written domains (Fig. 2h). Subsequently, we utilized H-KPFM to establish a ground truth image of the CPD, as depicted Fig. 2i. We conducted SS-KPFM with a constant 10 spiral cycles, adjusting the frame rate to 0.5 fps, 1 fps, 2 fps, and 3.3 fps, as shown in Fig. 2j–m, respectively. The findings demonstrate that SS-KPFM can recover the CPD of the pattern with high quantitative accuracy while maintaining high spatial resolution. Notably, our results so far highlight a significant breakthrough in KPFM imaging speed, with SS-KPFM operating at a rate of ~3.3 fps, while preserving high spatial resolution

and accuracy as shown by the profiles in Fig. 2n. This imaging speed sets a benchmark in KPFM imaging, surpassing the previously reported state-of-the-art[35] imaging speed by 2 orders of magnitude in a sample with similar topographic features (see topography in Supplementary Fig. 6). This advancement opens possibilities to investigate nanoscale charge dynamics that were previously obscured, with deeper insights into material behaviors and device mechanisms.

## Electrochemically mediated ion diffusion on LAO/STO

As previously highlighted, SS-KPFM's rapid recovery of spatiotemporal dynamics makes it a valuable tool for studying charge transport and electrochemical processes in dynamic systems and operandi devices. To demonstrate this, the charging and discharging processes of an LAO/STO device is investigated within a planar capacitor electrode geometry (Fig. 3a). LAO/STO is of particular interest due to its potential applications in electronics and spintronics, with the formation and manipulation of a 2D electron gas (2DEG) at the buried heterointerface offering possibilities for modern electronic devices[55]. However, for practical implementation, a comprehensive study of surface charges (oxygen vacancies and surface-adsorbed species) on the LAO surface is critical, as they are known to significantly influence the electronic properties, conductivity behavior, and potential applications of the interface in devices and quantum phenomena research[56].

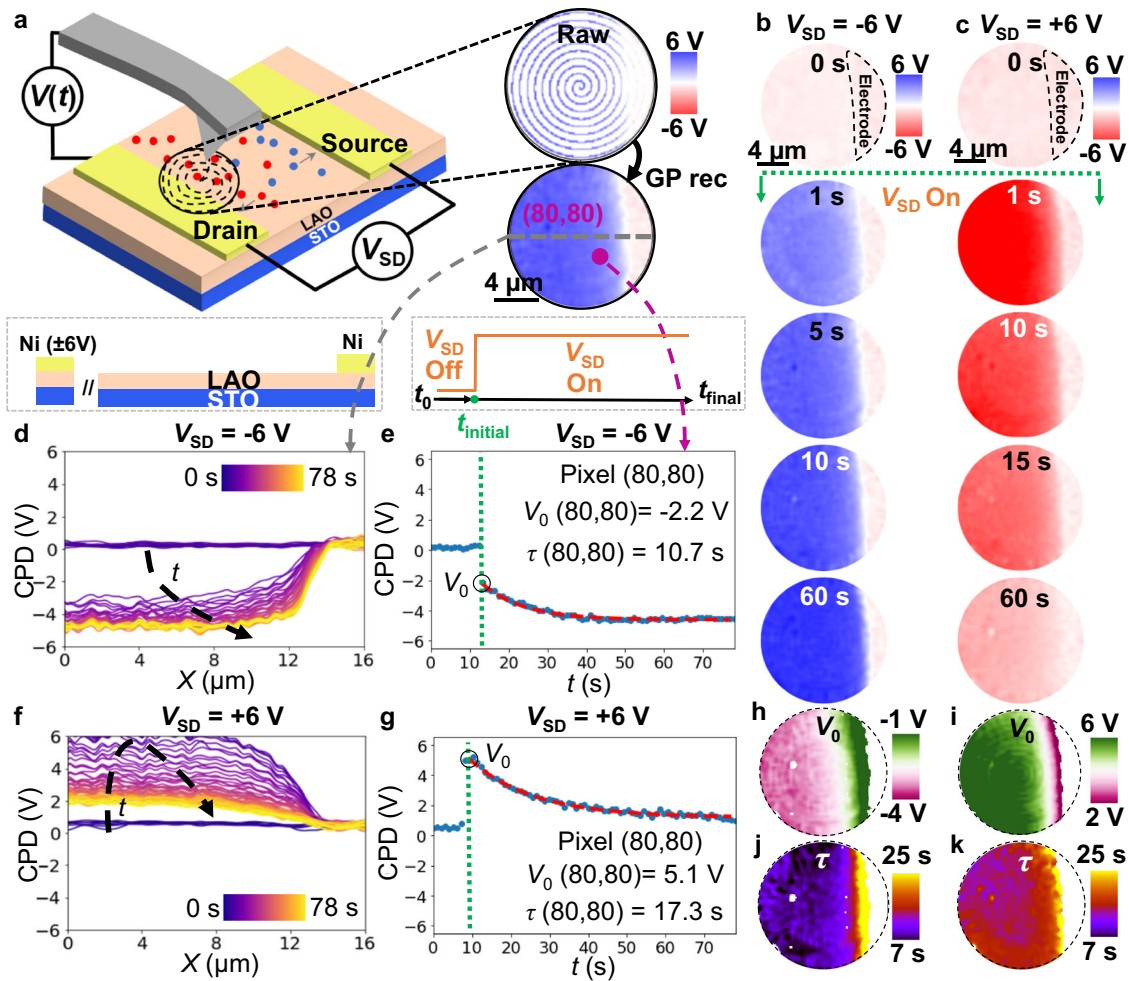

**Fig. 3 | SS-KPFM on LAO/STO lateral device. a** Schematics of the experimental setup, Raw data for a single CPD frame and reconstructed CPD frame using GP reconstruction. **b, c** CPD maps of LAO/STO—and part of the grounded electrode—during negative and positive biasing of the opposite electrode ($V_{SD} = -6$ V for (**b**) and $V_{SD} = +6$ V for (**c**)) respectively. **d, f** are CPD profiles over time during the first 78 s (rainbow color scale) after excitation for $V_{SD} = -6$ V and $V_{SD} = +6$ V, extracted

from sections of all the CPD frames along the gray dashed lines in (**a**). **e, g** are CPD evolution over time for a single pixel (marked in purple: number 80,80). Green dashed line indicates the time for the application of the $V_{SD}$ bias. Red dashed line shows the exponential decay fit. Black circle indicates $V_0$. **h, i** are the $V_0$ maps and **j** and **k** are the $\tau$ maps extracted from the fits performed point by point. **j, k** are the corresponding $\tau$ maps.

In this study, the LAO layer is 3-unit cells thick, ensuring that the heterostructure remains insulating, with no 2D electron gas (2DEG) present at the LAO/STO interface[57]. CPD spatiotemporal maps were captured after the application of ±6 V to the source electrode for ~80 s, providing information on the charge dynamics near the capacitor electrode. The images are captured at a rate of 1 fps and stored for further analysis allowing the observation of the charge and discharge processes in more detail (see full sequence in Supplementary Movies 1 and 2). Subsets of the image frames, shown in Fig. 3b, c, taken just before the excitation (0 s) and at 1, 5, 10, 15, and 60 s after the excitation, demonstrate the spatiotemporal dynamics of the system. Figure 3d, f depicts the time evolution of the CPD profiles along the gray dashed lines as a function of the distance from the grounded electrode. From the single pixel CPD dynamics in Fig. 3e, g, two distinct time regimes can be observed: an initial process faster than we can capture by SS-KPFM, likely occurring in the ms regime[58], followed by a slower process that takes tens of s to fully relax. Based on the distinct fast and slow timescales observed, we propose that these processes are associated with activation of single or mixed ionic subsystems during the charge and discharge process. In contrast to simple dielectric-based capacitors, where voltage drops homogeneously across their thickness, the observed dynamics and space distribution of CPD suggest the presence of an oxide with an ion-conducting surface electrolyte layer sandwiched between the two-metal electrodes. This scenario is further supported by macroscale device testing, where bulk impedance measurements were conducted on the same device (see Supplementary Fig. 7), revealing a broad distribution of characteristic time constants which are likely associated with the migration and diffusion of mobile surface ions in the presence of an electric field.

Quantitative analysis was performed by fitting the data to an exponential decay at every single pixel (red dashed line in Fig. 3e, g), finding the initial jump, $V_0$ and the characteristic time of relaxation, $\tau$. The following exponential decay function has been fitted:

$$V_{CPD}(t) = \frac{V_0}{2}\left(1 + e^{\frac{-t}{\tau}}\right) \tag{1}$$

where $V_{CPD}$ is the measured CPD, $V_0$ is an initial CPD value after the application of voltage, $t$ is the elapsed time since the application of the supply voltage and $\tau$ is the time constant associated to the slow formation of the depletion layer next to the capacitor drain electrode. By fitting the transient behavior at each pixel, we generated $V_0$ and $\tau$ maps displayed in Fig. 3h–k. These maps facilitate the clear visualization of local ionic charge carrier movement across the sample, particularly at areas of local heterogeneity.

As anticipated, SS-KPFM is limited in its sensitivity to relaxation processes with characteristic times faster than the imaging frame rate. Previous studies, utilizing time-resolved KPFM, have shown that the same devices exhibit relaxation processes on the order of tens of milliseconds, which were postulated to be associated with the transport of H+ and OH− ions through the full length of the channel[59]. Relaxation processes on similar timescales are also evident from impedance spectroscopy measurements (Supplementary Fig. 7). However, in SS-KPFM, our focus lies within the slower redistribution of charges with a time frame of tens of seconds. We propose that this slower relaxation is governed by water splitting at the electrode interface[60,61]. The evidence supporting the electrochemical mediation of slow ion diffusion is multifaceted. First, CPD measurements as a function of the lateral DC bias step (shown in Supplementary Fig. 8) reveal that the slower time response is only activated for biases higher than 250 mV, suggesting the presence of an activation energy barrier needed to trigger such process. Second, we observe an asymmetry in the dynamics for positive and negative electrode polarization which can be attributed to the primary carriers responsible for driving the internal polarization within the capacitor. Third, once the system is relaxed, the voltage drop accumulates right at the electrode/channel interface, indicating that the limiting process could be related to charge injection at this interface, due to the water splitting. To support this idea, bulk impedance measurements were conducted as a function of the environment relative humidity (RH), see Supplementary Fig. 9. We observed that the "fast" characteristic times slow down as the relative humidity increases. This suggests that the higher water presence on the surface promotes enhanced migration of H+ and OH− ions. Conversely, the "slow" characteristic times, as indicated by the impedance data and consistent with timescales seen in SS-KPFM, appear to diminish under lower relative humidity conditions, underscoring the significance of water in the electrochemical surface reactions that take place.

In summary, integrating macroscale and nanoscale measurements through SS-KPFM offers significant insights into the charge dynamics of operando devices. For our LAO/STO devices, the fast processes are related to the migration/diffusion of H+ and OH− groups and can be well-captured by tr-KPFM[59], while the slower voltage-dependent relaxation on the order of 10 s of seconds is linked to the electrochemical process of water splitting, and can be mapped spatiotemporally through SS-KPFM. As such, SS-KPFM's ability to provide nanoscale spatial information and rapid CPD maps in a dynamic system supplements the capabilities of existing tr-KPFM techniques, enabling a more comprehensive understanding of charge transport and electrochemical processes. The combined use of SS-KPFM and macroscale device testing will prove instrumental in unraveling the complex charge dynamics at various time scales, paving the way for improved device design and utilization in electronic and other related applications.

## Mapping of oxygen vacancy diffusion in polycrystalline TiO₂

In our prior experiment, we concentrated on exploring the alterations in charge dynamics on an operando device governed by mobile surface ions. However, for non-device systems, it is possible to utilize the strong electric fields generated by the small radius of the probe to locally modify nanoscale volumes of a material induced by the tip. This feature enabled us to leverage SS-KPFM for subsequent investigations of charge relaxation processes in a nanoconfined region of a polycrystalline $TiO_2$ thin film. Although conventional KPFM has been effective in investigating charge diffusion in various types of materials with characteristic times ranging from min to h or even days[62–65], materials with fast diffusion rates can undergo processes within fractions of a second, which are too rapid for conventional KPFM to capture. Nevertheless, as previously demonstrated, SS-KPFM can provide information at these faster timescales.

$TiO_2$ is a multifunctional semiconductor metal oxide extensively utilized in applications ranging from photocatalysis[66] and pigmentation to sunscreens and solar cells. Its optical, electronic, and photocatalytic properties are closely intertwined with its defect structure, particularly oxygen vacancies ($V_O$). These vacancies influence its light absorption, charge transport, and surface reactivity. Gaining insights into the local dynamics of $V_O$ is pivotal, as they play a critical role in determining the material's overall performance.

The experiment is performed as follows: First, the tip is brought into contact with the $TiO_2$ surface at the center of the scanning region, and a voltage pulse of varying amplitude and duration is applied (Fig. 4a). After the pulse is complete, SS-KPFM is triggered (Fig. 4b) and begins to simultaneously and non-invasively interrogate both structure (Fig. 4c) and CPD (Fig. 4d) of the charged region. In the topography channel (Fig. 4c), we can visualize the different $TiO_2$ grains of the sample and observe no topographical changes in response to the applied field. In Fig. 4d, the initial CPD maps ($t = 0$ s) exhibit confinement of the charged region, and the subsequent frames show the relaxation of the localized charges taking place over tens of seconds, gradually returning to the initial surface charge state of the pristine sample.

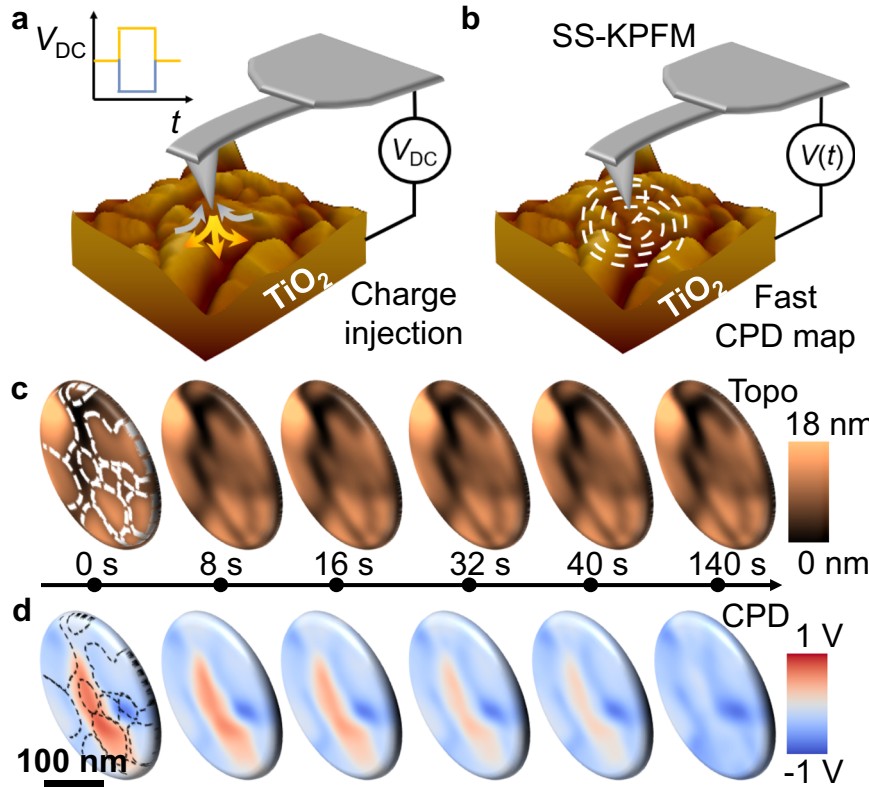

**Fig. 4 | Mapping oxygen vacancy diffusion in polycrystalline TiO₂. a** A bias pulse of variable duration and amplitude is applied to the tip while in contact. **b** Right after the pulse is complete, SS-KPFM is triggered, and surface charge is mapped at 2 fps. **c** Selected topography maps after the application of 10 V during 25 s at different times after pulse, namely (0, 8, 16, 32, 40, and 140 s). **d** Corresponding CPD maps at the same times.

The dynamics of negatively charged oxygen ions (O⁻²) and positively charged oxygen vacancies are frequently discussed in the context of oxide-based switching devices[67,68]. The movement of O⁻² leads to the redistribution of $V_O$ within the oxide material, and conversely, the diffusion of $V_O$ can be seen as a consequence of the O⁻² migration in the opposite direction. Both the migration of O⁻² through defect sites and the diffusion of $V_O$ are mechanisms often explored in the literature concerning resistive switching in TiO₂[67,68].

In the Supplementary Fig. 10, we examine the systems behavior under the application of bipolar bias pulses, revealing a clear asymmetry in the CPD dynamics between application of positive (Supplementary Fig. 10e–g) or negative biases (Supplementary Fig. 10h–j). Animations with all the CPD frames are provided in Supplementary Movies 3–8. The data indicates that when a positive voltage is applied, there is a corresponding positive change in the CPD, and vice versa for negative voltages. Furthermore, the relaxation dynamics exhibit quicker responses under positive bias and more gradual adjustments under negative bias. These results are in agreement with related SPM measurements on similar TiO₂ film which indicated that the sign of the applied bias governs the charge injection/redistribution mechanism with $V_O$ being formed under a positive bias resulting into a more conductive state and electrons being injected into oxygen vacancies (possibly from another O²⁻ ion) under negative bias thus oxidating the sample and turning it into a more insulating state[67].

This approach can also be useful for probing charge injection/redistribution dynamics spatially across the sample surface, to visualize how transport is impacted by grain boundaries and other sample heterogeneities. As a visual representation, Supplementary Fig. 12 displays the same experiment, but with the local field application executed in varying locations. Specifically, the experiment is conducted independently on two adjacent grains and on the grain boundary, where the tip simultaneously contacts both grains.

Continuing our investigation, we delved into the temperature dependence of the charge screening process. To achieve this, we applied a local field (−8 *V* bias pulse for a duration of 1 min) and subsequently performed SS-KPFM charge relaxation tracking on the same grain, at various temperatures (Fig. 5). Employing this experimental method gave us invaluable understanding into the influence of temperature on charge screening dynamics, offering essential information on the charge transport mechanisms within the polycrystalline TiO₂ thin film. In Fig. 5a–f, the impact of temperature on the charge diffusion process is clear, with subsequent charge relaxation towards the pristine state occurring faster at higher temperatures and slower at lower temperatures. The averaged ΔCPD (ΔCPD = CPD − $CPD_{t=120\,s}$) evolution (Fig. 5g) over the central region (colored dashed squares in Fig. 5a, c, e) represents the mean response of the center of the grain for each different temperature, where the temperature evolution can be clearly observed. The relaxation behavior followed an Arrhenius-type law ($D = D_0 e^{\frac{-E_A}{K_B T}}$, where $D$ is the diffusion coefficient, and $E_A$ is the associated activation energy, $T$ is the temperature and $K_B$ is the Boltzmann constant) allowing extraction of the characteristic diffusion time derived from the SS-KPFM results (Fig. 5h). This fitting procedure yielded a value for $E_A$ of the charge diffusion process, which was found to be 0.18 eV.

As previously stated, it is well-known that $V_O$ routinely forms on TiO₂ surfaces[69], and their diffusion can drive surface reactions and photoinduced surface dynamics[70]. Our measured activation energy is similar to the diffusion barrier predicted by first principles, calculated at 0 K (0.35 eV)[71]. The somewhat lower value measured by our experiments may be explained by the presence of an external electric field, not considered in the simulations, and which would lead to surface charging and altered lattice strain caused by local geometry reorganization. Density functional theory (DFT) studies have shown that the formation energy of $V_O$ is highly sensitive to changes in local charge environments and lattice strain[72–75], and we therefore use DFT

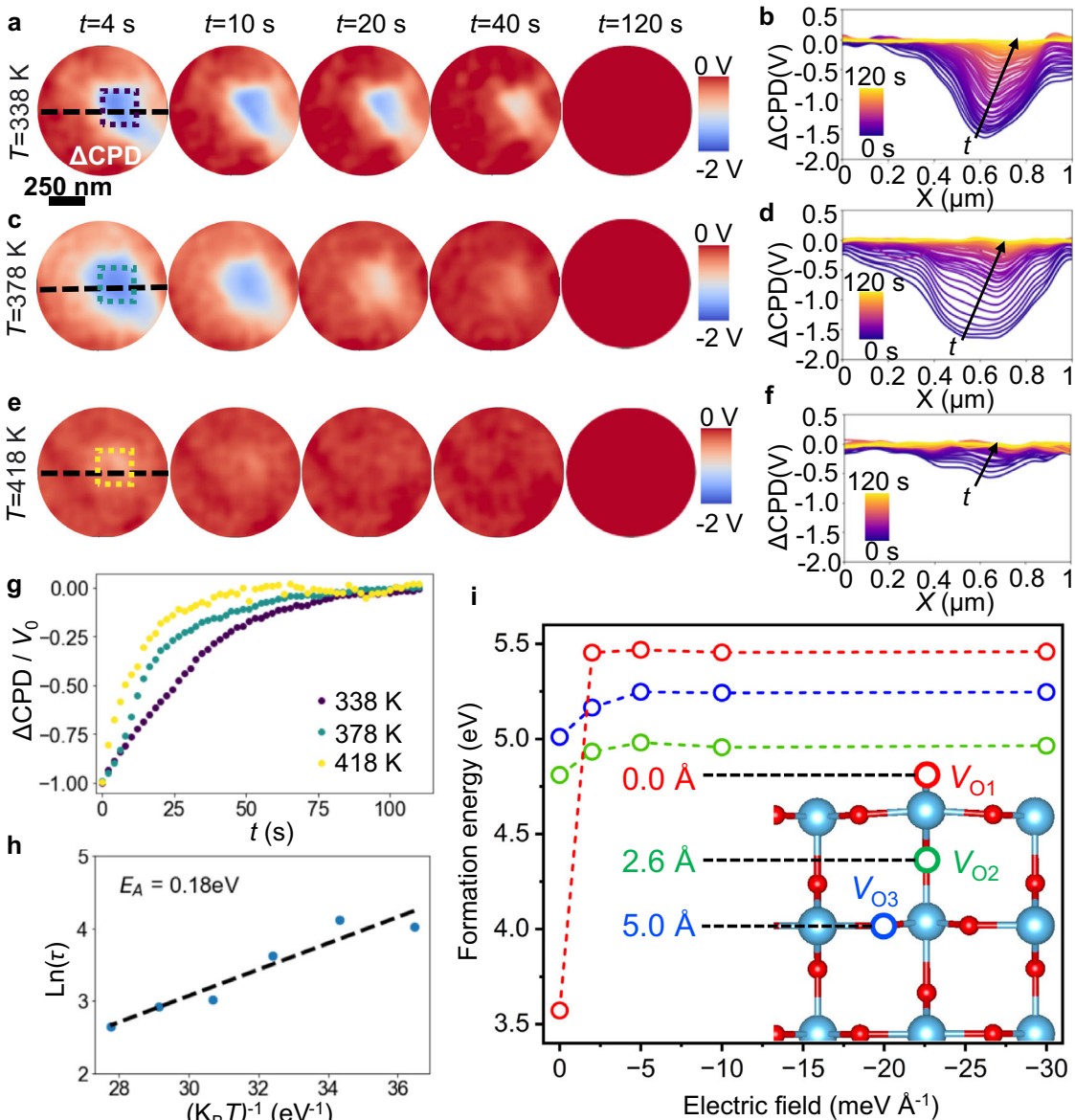

**Fig. 5 | Activation energy and temperature dependent oxygen vacancy diffusion.** ΔCPD maps after application of −8 V during 1 min at different times and for different temperatures: **a** 338 K, **c** 378 K and **e** 418 K. **b**, **d**, **f** Corresponding profiles along the dashed black line over time. **g** ΔCPD evolution averaged over the central region (dashed squares in (**a**) at 3 different temperatures. **h** Arrhenius-type fit to obtain $E_A$. **i** DFT calculated external electric field-dependent formation energy of oxygen vacancies on a (110) $TiO_2$ rutile surface. The surface vacancy ($V_{O1}$, or 0.0 Å) is marked in red, nearest subsurface vacancy ($V_{O2}$, or 2.6 Å) in green, and deeper vacancy ($V_{O3}$, or 5.0 Å) in blue. The distance reported is distance from the surface.

calculations in the Vienna ab initio simulation package (VASP)[76,77] to explore the possibility of external electric fields reducing the $V_O$ diffusion barrier in our experiments.

The formation energy for an oxygen vacancy on a (110) surface ($V_{O1}$, or 0 Å below the surface) was calculated at the GGA + U-level and compared with the energies of a subsurface vacancy 2 layers (≈2.6 Å) below the surface ($V_{O2}$), and a subsurface vacancy 4 layers (≈5.0 Å) below the surface ($V_{O3}$, Fig. 5i). The Hubbard U parameter for our calculations was 4.2 eV, which has been well-established to correctly describe the electronic structure, Ti 3d orbitals, and reaction energies for rutile $TiO_2$. While the energy of the most stable vacancy ($V_{O1}$) can vary due to differences in calculated supercell and surface sizes, our predicted value of 3.58 eV for a (110) surface is similar to other studies (≈3.6 to ≈3.7 eV) using GGA + U[78–80]. As expected with $TiO_2$, the formation energy increases as oxygen vacancies are moved deeper into the material without the addition of an electric field. If we apply a

negative external electric field along the c axis (Fig. 5e), the formation energy increases for all vacancies, but the effect is far more drastic at the surface than at the subsurface. The increase in relative energy of the surface vacancy indicates that the negative electric field in our experiments would likely lower the diffusion barrier for oxygen vacancies by promoting their migration from the surface deeper into the subsurface of rutile $TiO_2$. Indeed, this prediction is consistent with many other experimental and theoretical findings such as measurements of insulating states caused by negative biases injecting electrons (e.g., from $O^{-2}$) into the defect site[67], scanning tunneling microscopy measurements demonstrating that sample bias from the tip alters the distribution of subsurface and surface $V_O$[81], the calculated >1 eV reduction of $V_O$ formation energy in p-type type interfaces versus n-type interfaces[82], and the DFT calculated external negative electric-field induced increase in the relative energy of $V_O$ on the surface versus the subsurface for the related $TiO_2$ anatase phase[83].

As discussed in the previous section, KPFM measurements in air environment may be affected by surface adsorbates, water layers and mobile surface ions, introducing additional complexities not fully captured in the DFT simulations. Such phenomena could act as a source of $O^{-2}$ either from the environment, surface electrochemistry or diffusing from other parts of the lattice to satisfy the vacancies needs. The implementation of SS-KPFM under fully controlled environment (glove box, liquid[84], or ultra-high vacuum[85]) will further enhance its potential to better understand such interfacial phenomena that likely impact materials behavior and device performance in-operando.

Overall, these results demonstrate that SS-KPFM can provide a comprehensive understanding of the charge screening processes in thin films or polycrystalline oxides. However, in some cases, more complex surface electrochemistry such as irreversible redox reactions mediated by water absorption or others can also be significant. Tracking these changes would benefit from temperature and humidity control[86] as well as additional correlative techniques such as in-situ XPS that lack nanoscale spatial resolution[62]. Although this is beyond the scope of this work, we anticipate that coupling SS-KPFM with environmental control such as humidity, temperature, and even photo-response would be a powerful approach for investigating surface and bulk screening characteristics. Finally, it is noteworthy that the experiment methodology described here (charge injection/redistribution by the AFM tip) would not be feasible using conventional time-resolved KPFM methods, and the relaxation dynamics are too rapid to be captured by the current benchmark in high-speed KPFM measurements[35].

## Conclusions

In this study, we introduce and implement SS-KPFM, which allows for high-speed spatiotemporal imaging of contact potential difference on the nanoscale at sub-second frame rates. We thoroughly investigate the technique's capabilities and limitations, including spatial and temporal resolution, quantitative accuracy, and Gaussian process-based reconstruction optimization. We demonstrate the capabilities of this method on technological relevant devices by successfully bridging nanoscale surface charge dynamics with macroscale device characterization on a LAO/STO device. Our findings reveal the substantial contribution of electrochemically mediated diffusion of mobile surface ions on timescales of tens of seconds. In addition, we further validate the approach in polycrystalline films, demonstrating its capability to induce and subsequently monitor ionic charge carriers, with our nanoscopic probe functioning both as a nano-electrode and sensor. This capability is demonstrated through the study of temperature-dependent oxygen vacancy motion in a polycrystalline $TiO_2$ sample at the single grain level. The results reveal Arrhenius-like processes, characterized by an activation energy of 0.18 eV, which are in good agreement with reported values of oxygen vacancy diffusion and further supported by DFT calculations. Overall, our results demonstrate that SS-KPFM sets a benchmark for high-speed KPFM, significantly outperforming existing methods with its sub-second frame rates. This breakthrough has far-reaching implications, with potential applications in surface science for energy materials, including ionic conductors, photovoltaics, and ferroelectrics. The advancement offered by SS-KPFM enables researchers to probe dynamic charge processes and gain deeper insights into the behavior and properties of low dimensional materials and miniaturized devices.

## Methods

### Experimental AFM

The spiral scanning AFM is achieved using a Cypher AFM (Asylum research an Oxford Instruments company) controlled via FPGA using a custom-built LabVIEW/python interface that drives the tip motion following a spiral trajectory while, H-KPFM is performed simultaneously using an external lock-in amplifier (Zurich Instruments, HF2LI). The limitations regarding the data acquisition of such

experimental scheme are given by the IO rate of the FPGA card, which in our case is $4\,MS\,s^{-1}$. The tips used are commercially available Multi75-G (BudgetSensors), with a free resonance frequency of 75 kHz and a spring constant of $3\,N\,m^{-1}$. The free amplitude used for the topography tracking was set to be 1 V of the photodiode signal ($\approx 100$ nm in displacement units) and the setpoint was set between 0.75 V and 0.8 V. The H-KPFM was performed, as explained in more detail in ref. 87, with AC frequency of $\omega_e = \omega_1 - \omega_0$, where $\omega_1$ and $\omega_0$ are the first and second cantilever eigenmodes respectively; for the tips used in this work $\omega_e \approx 400$ kHz, with small changes depending on each specific tip, and a bias amplitude of 3 V. The AFM experiments were performed in air at room temperature (25 °C), except for the temperature-dependent measurements of Fig. 5, where a temperature control sample holder was used. For the temperature-controlled experiments, the system was left for 20 min every time the temperature was changed to let the system thermalize and avoid drift.

### WS₂ on silicon

Monolayer $WS_2$ flakes were directly synthesized on a highly p-doped Si substrate using chemical vapor deposition (CVD) at atmospheric pressure. The aqueous precursor solution was coated onto the Si substrate by spin-casting at 3000 rpm for 1 min where the precursor solution contains ammonium metatungstate hydrate (Sigma-Aldrich, 463922), sodium hydroxide (Sigma-Aldrich, 306576), and iodixanol (Sigma-Aldrich, D1556). Then, the precursor-coated substrate and sulfur powder (Sigma-Aldrich) were placed at two different positions in the CVD chamber and heated to 750 and 210 °C, respectively for 8 min with introducing $500\,cm^3\,min^{-1}$ of $N_2$ and $5\,cm^3\,min^{-1}$ of $H_2$ gases at standard temperature and pressure. Additional details of the growth conditions and further characterization can be found in reference[88].

### PZT sample

$Pb(Zr_{0.2}Ti_{0.8})O_3$ (PZT) films were prepared on $LaAlO_3$ (LAO) 100 substrates with a bottom electrode of $La_{0.7}Sr_{0.3}MnO_3$ (LSMO) by pulsed laser deposition (PLD). 20 nm LSMO layers were deposited at 800 °C with an oxygen pressure of 120 mTorr. Following this 20 nm thick PZT films were grown at 700 °C and 115 mTorr of oxygen. The samples were cooled at 450 mTorr of oxygen at a speed of 20 °C min⁻¹. XRD reciprocal space mapping (RSM) and additional PFM characterization of identically prepared films can be found in ref. 89. For images in Fig. 2i, a checkerboard-like pattern of polarization was written into the PZT film by applying ±8 V with the AFM tip across the sample and subsequently performing single frequency PFM to observe the written pattern, before the H-KPFM experiments.

### LAO/STO lateral device

Details about the growth conditions for the LAO/STO sample and related defect structures of the specific films studied here are following procedure described elsewhere[90–92]. Briefly, STO substrates were etched with a chemical solution of ammonium bifluoride and hydrofluoric acid at pH = 6 to obtain a $TiO_2$-terminated surface and then annealed at 950 °C for one h in an oxygen- rich atmosphere. LAO films were grown on the cleaned STO (001) substrates by Pulsed-Laser Deposition (PLD) at a base chamber pressure of 10 − 6 Torr which was increased to an $O_2$ partial pressure of 10 − 4 Torr via an MKS Mass Flow Controller and Cold Cathode. The growth was performed at a temperature of about 750 °C with an initial ramping rate of about 10 °C min⁻¹ up to 500 °C and then about 30 °C min⁻¹ up to the deposition temperature. The LAO target was ablated using a 248 nm KrF excimer (Coherent Inc.) laser with a fluence of about 1.2 J cm⁻² and a repetition rate of 2 Hz. After deposition, films were brought to room temperature at cooling rates of about 10 °C min⁻¹ then about 5 °C min⁻¹. 3-unit cells thick films were grown, and the growth rate was followed in situ by oscillations in Reflection High-Energy Electron Diffraction (RHEED) patterns (STAIB Instruments). For electrical

measurements two Ni electrodes (≈50 nm) were evaporated onto the LAO surface with a channel distance of 40 μm. Additional characterization including X-ray diffraction analysis (XRD), reciprocal lattice mapping (RLM), high-angular annular dark-field (HAADF), electron energy loss spectroscopy (EELS) and medium energy ion scattering (MEIS) of the samples can be found in references[90–92].

## TiO$_2$ synthesis

A 65 nm thick TiO$_2$ thin film was deposited onto a Pt/Ti/SiO$_2$/Si substrate using a radio frequency (RF) magnetron sputtering system (Ultech Co., Korea). The TiO$_2$ target, with 99.99% purity, was used for the deposition, and argon gas was employed as the working gas at a flow rate of 30 cm$^3$ min$^{-1}$ at standard temperature and pressure. The sputtering process was conducted for 30 min at a sputtering power of 180 W, pressure of $4 \times 10^{-6}$ Torr, and a temperature of 350 °C. Subsequently, the TiO$_2$ thin film was annealed in a furnace at 950 °C for 1 h to achieve the desired properties. Structural characterization is shown in Supplementary Fig. 13 with SEM images and XRD patterns.

## DFT calculations

DFT calculations used the VASP[76,77]. The Perdew-Burke-Ernzerhof (PBE) functional was used with a Hubbard U correction of 4.2 eV, which is well-established to accurately describe vacancies in rutile TiO$_2$[78–80]. The lattice constants were calculated at the unit-cell level using an electronic energy cut-off of 450 eV, and with the electronic and force convergence criteria set to $10^{-6}$ eV and 0.01 eV Å$^{-1}$, respectively. A Monkhorst-Pack K-Point mesh was used for bulk unit-cell calculations (8x8x8). Surfaces were constructed by expanding the unit-cell into a larger supercell and cutting the $4 \times 2$ TiO$_2$ (110) surface from the bulk (64 TiO$_2$ units, or 192 atoms). The calculated bulk unit-cell lattice parameters were 4.67 for $a$ and $b$, and 3.03 for $c$, which agrees well with other studies[93]. The (110) surface added a vacuum spacing of 15 Å along the $c$ axis to avoid spurious interactions between slabs. The $a$ and $b$ lattice constants for the surface were 12.15 and 13.23 Å, respectively. All surface calculations used the same energy cut-off as the bulk calculations, but the K-Points were changed to the Γ-point, and the convergence criteria to $10^{-5}$ eV and 0.02 eV Å$^{-1}$ for the electronic self-consistency loop and the forces, respectively. All surface calculations mimicked the bulk by freezing the bottom-most three layers, and an electrostatic potential was added along the c axis for all E-field calculations. The formation energy was calculated using the expression $E_\mathrm{f} = E_\mathrm{def} - E_\mathrm{defect\text{-}free} + \mu_0$ where $E_\mathrm{def}$ and $E_\mathrm{def\text{-}free}$ are the total energies of the defective and non-defective slabs at their respective electric field, and $\mu_0$ is the chemical potential of an oxygen atom.

## Data availability

The data that support the findings of this study are available from the corresponding author upon request.

## Code availability

Python scripts used for the analysis are available in Github (https://github.com/mchecanu/Spiral-scanning-KPFM) and Zenodo[94].

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

## Acknowledgements

This work was supported by the Center for Nanophase Materials Sciences (CNMS), which is a US Department of Energy, Office of Science User Facility at Oak Ridge National Laboratory. This manuscript has been authored by UT-Battelle, LLC, under Contract No. DEAC05000R22725 with the U.S. Department of Energy. S.Y. acknowledges support by the U.S. Department of Energy, Office of Science, Basic Energy Sciences (BES), Materials Sciences and Engineering Division. Y.K. acknowledges support from the National Research Foundation of Korea (NRF) grant funded by the Korea government (MSIT) (No. NRF-2021R1A2C2009642). A.S. acknowledges support from the Air Force Office of Scientific Research (FA9550-21-1-0005). P.S. acknowledges ORNL's CNMS User Nanoscience Research Program (CNMS2021-B-00849) and Flinders University Start-up grant. We gratefully acknowledge Dr. J. Y. Wang (Northwestern Polytechnical University) for providing the PZT samples.

## Author contributions

M.C. and L.C. conceived and designed the research. M.C. performed the experimental measurements. A.F. performed the TiO2 DFT calculations. R.V. and M.Z. built the image reconstruction routines. S.Y. and K.X. grew the WS2 sample. A.S. designed and built the LAO/STO devices. C.S. and Y.K. synthetized and characterized the $TiO_2$ sample. S.J. built the LabVIEW script to control the AFM. S.J., M.C., and K.K. wrote the python script to control the FPGA. I.I. performed the bulk impedance measurements. P.S. provided initial PFM results and analysis on PZT samples. N.D. assisted with data analysis and interpretation. M.C. and L.C. wrote the manuscript with the assistance of all co-authors.

## Competing interests

The authors declare no competing interests.
