## [Peer Review File · Nature Communications]

High-speed mapping of surface charge dynamics using sparse scanning Kelvin probe force microscopyReviewers' Comments:

Reviewer #1:

Remarks to the Author:

The manuscript of Checa et al. entitled "High speed mapping of surface charge dynamics via Spiral Scanning Kelvin Probe Microscopy" is of highest technical quality and in my opinion should be definitely published. The authors demonstrate that the application of spiral scanning in combination with suitable intelligent algorithms allow to overcome fundamental problems in time resolved Kelvin Probe Microscopy. The technical implementation and the chosen applications are absolutely the technical forefront! Nevertheless, the question arises, if the combination of this two previously reported techniques is sufficient novel for publication in Nature Communications.

This reviewer has some doubts in respect to the novelty. The authors omit the first papers on spiral scan (sometimes also referred to as Lissajous scanning) which date back to 2009 / 2010 ("Fast spiral-scan atomic force microscopy" 10.1088/0957-4484/20/36/365503 and "High-speed cycloid-scan atomic force microscopy" 10.1088/0957-4484/21/36/365503 both in Nanotechnology; both papers received about 100 citations !). The concept of applying this scan path has been implemented also for Scanning Electrochemical Microscopy ("High-Speed Electrochemical Imaging" 10.1021/acsnano.5b02792 in ACSNano 2015!). The work of Ando and coworkers for biological samples should be also mentioned. The work of Ziegel and Ashby is very important but by far not the only nor the first one. The important AI-based algorithms of the here-presented paper have been published previously in Small.

In summary, this reviewer sees the novelty of the present approach in combining existing techniques, albeit on the highest level. However, taking this point into consideration a more specialised journal would be more appropriate as the manuscript is centred around the technique and not the materials. The manuscript is otherwise excellent and merits definitely publication!

Reviewer #2:

Remarks to the Author:

The work entitled "High speed mapping of surface charge dynamics via Spiral Scanning Kelvin Probe Force Microscopy" by Checa et al. nicely renders the importance of resolving dynamic processes not only in terms of topographical changes, but also as a function of their electronic properties in time and length scales.

Nevertheless, a few things should be updated.

1. "These techniques are known to provide valuable insights into surface charges, dielectric properties, and the contact potential difference (CPD), which represents the difference in work function between the tip and the sample." After this section, I recommend to include two sentences, which are focused on the achievements of KPFM at the atomic scale. Here I would recommend the following citations:

"Gross, L.; Mohn, F.; Liljeroth, P.; Repp, J.; Giessibl, F. J. & Meyer, G. Measuring the Charge State of an Adatom with Noncontact Atomic Force Microscopy Science, 2009, 324, 1428-1431",

"Mohn, F.; Gross, L.; Moll, N. & Meyer, G. Imaging the charge distribution within a single molecule Nature Nanotechnology, Nature Publishing Group, 2012, 7, 227-231", and

"Fatayer, S.; Albrecht, F.; Zhang, Y.; Urbonas, D.; Pena, D.; Moll, N. & Gross, L. Molecular structure elucidation with charge-state control Science, Science, American Association for the Advancement of Science, 2019, 365, 142-145".

This would allow to reader to learn about the capabilities of modern KPFM.

2. "The bandwidth capabilities of H-KPFM were previously investigated by Garret et al., who reported a remarkable improvement in imaging speed of more than 100-fold compared to conventional FM-KPFM." I would prefer to have a repetition of the citation directly behind "Garret et al."

3. "From each sparsely sampled region, ..."

Can the authors elaborate more on the term "sparsed sampling". In my understanding, sparsed sampling includes the a priori knowledge of certain image properties. The spiral however, samples not the complete real space without the knowledge of the "feature" distribution etc. In my opinion, this is a reduced sampling, not a sparse sampling.

4. "... controlled via FPGA using a custom-built LabVIEW/python interface that drives the tip ..." What is the speed limit for this software approach?

5. "... The image was acquired using a tip trajectory of a spiral with 24 internal cycles and an imaging speed of 4 seconds per frame. ..." Have more scan geometries been tested, e.g. smaller pitch sizes / more cycles per frame?

6. The authors should include some more of their experimental parameters and conditions. I miss the resonance frequencies of the cantilevers, the oscillation amplitude in nm used for imaging. Also the types and manufacturers of the used cantilevers does not seem to be given.

If the comments above are taken into account, then I truly recommend this work for publication. It is an interesting approach to push the limits of scanning probe microscopy towards time depended material contrast observations.

Other than that I really like the presented work.

Reviewer #3:

Remarks to the Author:

In this contribution, a new mode of Kelvin probe force microscopy is introduced. Principles of the method and differences to existing modes are explained well and advances beyond the state-of-the-art are clearly shown. These advances are highly relevant to the field as shown in some characteristic examples from research on materials for electronic or energy applications. A remarkable impact of this work can be expected and I support a publication in Nature Communications.

Some aspects listed in the comments below, however, may deserve the authors' attention to sharpen the focus of the work and increase readability for a widely interested readership. Some of the mechanistic explanations appear quite speculative and handwaving. Such details may either be omitted or, in case they are regarded relevant, be explained in a more explicit way. Therefore, a revision is recommended:

1. The graphical abstract Figure is very busy, although it very nicely summarizes the work. It would help to use different color codes for CPD and time. Further, the meaning of the vertical axis stays unclear. I would probably omit the whole plot of "time dynamics" (anyway "space" would be the appropriate axis assignment) and the two V_0 and τ maps to focus on the central message: spiral scanning plus GP in-painting leads to fast mapping of CPD.

2. It should be considered throughout the text that the term "dynamics" by definition contains a temporal change. Therefore, an individual CPD map cannot contain any dynamics, just a time-series of such maps does (see, e.g., last sentence of abstract).

3. It would be useful to indicate that the minimum time to measure a KPFM image of 16s mentioned on the top of page 2 given by the limitation of the topography feedback loop would be considerably shorter for planar samples. Reference to this thought would also be relevant on the bottom of page 4, where the analysis of planar samples is discussed.

4. The time scale should be given in a unified way. Either the number of frames per second, or the time needed for a measurement or the time resolution of a physical quantity (e.g. CPD) should be used. This will help to compare different statements in the manuscript and to follow comparison to references.

5. In the direct comparison of Figure 1, the raster method seems to provide the superior spatial resolution. It should be indicated in the Figure or the legend, that the temporal resolution is far better in the spiral scan approach.

Also in Figure 2, the time needed for the raster image would be nice to see. Perhaps a) the time needed to measure the same area as that covered by the spiral or b) the time needed to measure ~15 "checkerboard fields" would provide a valid comparison.

6. The environment for the measurements (air, inert gas, vacuum?) needs clear specification.

Otherwise, explanations about water layers, ionic movements etc. lack relevance.

7. Assignments of figure parts and legend are not consistent in Figure 3. Further, in the experimental part Ni is mentioned, Au is indicated in the Figure.

It would help if the position 80 would also be shown in parts d) and f), or, even better, if the pixel chosen for e) and g) would be positioned on the line for which data are plotted in d) and f). The text "j) and k) ... point by point." needs to be omitted.

8. On page 9, it would help to clarify which electrode is referred to as source/drain.

9. The explanations provided from the bottom of page 10 either should be omitted (if the focus is just placed on the new KPFM method) or need more specific reference to prior literature on LAO (if the reader should have a chance to follow the authors' argument).

10. The meaning of the results discussed on page 11 strongly depend on the medium the films are in contact with (see comment 6). For electrochemical impedance measurements, a liquid contact is typically used. This is not mentioned in the text. Diffusion of ions is also compared to diffusion coefficients of ions in (diluted) aqueous solutions. Is there a water film present for which a similar role can be assumed? Otherwise, ions bound to the oxide surface would have to be discussed. Was there any influence of humidity studied? Is there a distinguished meaning of positive or negative polarization against an external reference (electrode?) or is it just meaningful for one electrode relative to the other? Similar: any meaning of grounding? Any chemical reactions possible under the given experimental conditions? A rather complex discussion is opened but not concluded to satisfaction. Again, it could either be omitted for the sake of focussing on the method and neglecting the aspects of materials science or the details needed to follow the discussion should be added. At present the role of different charge carriers (electrons, anions, cations) remains unclear and the readership has no chance to develop a clear understanding which "faradaic processes" or "surface electrochemical activity" is possible under the given circumstances. At present, the arguments stay handwaving. If the authors have a clear working hypothesis backed up by the literature they are welcome to explicitly present it. Otherwise, it may be better to leave such analysis to detailed future measurements. Such explanation at least is not mandatory at present to show the benefits of spiral scanning in KPFM. If understood correctly, the voltage drop is maximum "in front of" or "close to" the more positive electrode, not "in" it.

11. Explanation of the TiO₂-results lack consistency. O₂ is not an ion. Negative species must be thought of to be driven into the lattice (e.g. O with the charge of -2?) at negative polarization or positive species pulled out (such as vacancies of negatively charged O-ions). To consider such vacancies being pulled out at positive bias appears counter-intuitive. Further, which evidence do the authors have for moving O-ions and vacancies as opposed to electrons, H⁺ or OH⁻?

In these experiments, it is not explained why a voltage pulse at 8V can lead to a higher CPD than a pulse at 10 V. A higher CPD indicates a higher concentration of charges for +8V. It is unclear, however, why such higher concentration would not diffuse across the grain boundary, but the smaller concentration at +10 V does. Rather, a barrier could be overcome at 10V and the process was too fast to resolve it in the present experiments.

It cannot be stated that the initial surface charge state is recovered within the measurement window since significantly different patterns are left after 140 s at either +10V or -10V. Subtraction of the line measured at 140s from all lines (background correction) would probably also show this very clearly.

12. Structural and elemental characterization of the sample materials are missing. If the authors have prepared the same materials previously under identical conditions (as it appears from the text) reference to such earlier work would suffice.

Reviewer #4:

Remarks to the Author:

The manuscript "High speed mapping of surface charge dynamics via Spiral Scanning Kelvin Probe Force Microscopy" by M. Checa et al. shows the capabilities of spiral SKPFM for high-speed acquisition surface electrical properties in nanoscale materials. The power of the technique is demonstrated on two highly interesting material systems resulting in interesting findings on the relaxation dynamics. Overall, the results are presented in an excellent way. The existing literature is reviewed well and the results are set into context. The article is well written and is certainly suitable for Nature Coms, however, a few aspects could be still improved:

While the authors show clearly that the performance of the technique could be significantly improved with respect to the previous reports by Garret et al. by introducing the spiral scanning (the supporting info shows images and performance data acquired in raster scanning and spiral scanning), the discussion in the last paragraph of the spiral scanning section on page 8 the technical factors that limit the speed of the technique are only discussed superficially. It would be nice to have here or in the supporting info a discussion on what are the actual technical limitations. Is it the FPGA card you used? Which are the limiting time constants here? Is it the output bandwidth of the lock in amplifier, or the feedback itself? Can you give experimental proof of this? Which would be the route for improving the speed further? I understand that it is not the aim to make a technical article out of this, but it is important to give a perspective for further technical development.

Concerning the section on the LAO/STO lateral capacitor: it is not fully clear if you measure the fast (40 and 80 ms) time constants in the diffusion processes. Please clarify this. Please also discuss a bit more in detail the comparison with impedance spectroscopy data. There is certainly a difference between macro and nanoscale measurements.

Reviewer #1:

The manuscript of Checa et al. entitled "High speed mapping of surface charge dynamics via Spiral Scanning Kelvin Probe Microscopy" is of highest technical quality and in my opinion should be published. The authors demonstrate that the application of spiral scanning in combination with suitable intelligent algorithms allow to overcome fundamental problems in time resolved Kelvin Probe Microscopy. The technical implementation and the chosen applications are absolutely the technical forefront!

Dear Reviewer,

We would like to express our sincere gratitude for taking the time to review our manuscript entitled "High speed mapping of surface charge dynamics via Spiral Scanning Kelvin Probe Microscopy" and providing such positive and encouraging feedback. We are thrilled that you recognize the high technical quality of our work and its potential for publication.

Your acknowledgment of our innovative application of spiral scanning in combination with intelligent algorithms to overcome fundamental problems in time-resolved Kelvin Probe Microscopy is much appreciated. We are glad that you found our work at the forefront of the field.

Nevertheless, the question arises, if the combination of these two previously reported techniques is sufficient novel for publication in Nature Communications.

We appreciate the reviewer's thorough evaluation. While we agree that Nature Communications upholds stringent standards for novelty, we firmly believe that our additional measurements, models, and interpretation presented in the newly submitted manuscript satisfy these criteria.

It is indeed true that we have combined multiple existing methods to develop a uniquely innovative approach. The significance lies in the collective outcome rather than the individual components. In its current form, our new method significantly enhances the imaging speed of KPFM by 2 orders of magnitude, presenting an exciting opportunity to investigate diverse nanoscale charge dynamic processes.

We also emphasize that the combination of disparate approaches to explore novel phenomena should not disqualify a work from publication in a high-impact journal like Nature Communications. In fact, this practice is often observed in esteemed publications, as demonstrated by the example of electrochemical strain microscopy (*Nature Nanotechnology* 5 (10), 749-754), which effectively extended the principles of PFM to investigate ion conductors, despite the absence of technical differentiation. Similarly, the application of EFM in liquid environments (**Nature Communications** 5.1 (2014): 3871) illustrates the integration of different methodologies to achieve novel outcomes. However, we can look at more recent works for compelling instances, such as super-resolution AFM, which borrowed principles from super-resolution fluorescence imaging and applied them to AFM. This amalgamation of techniques yielded remarkable results (*Nature* 594, 385–390 (2021)). Equally, the correlation of two existing approaches (super-resolution optical fluctuation imaging and Scanning ion conductance microscopy) have all been published in nature level journals (*Nature Communications*, 12(1), 4565.). Consequently, we firmly believe that our updated manuscript warrants publication in Nature Communications, as it aligns with established precedents.

Below we present a detailed point-by-point response to his/her comments.

- 1. The authors omit the first papers on spiral scan (sometimes also referred to as Lissajous scanning) which date back to 2009 / 2010 ("Fast spiral-scan atomic force microscopy" 10.1088/0957-4484/20/36/365503 and "High-speed cycloid-scan atomic force microscopy" 10.1088/0957-4484/21/36/365503 both in Nanotechnology; both papers received about 100 citations!). The**

concept of applying this scan path has been implemented also for Scanning Electrochemical Microscopy ("High-Speed Electrochemical Imaging" 10.1021/acsnano.5b02792 in ACS Nano 2015!). The work of Ando and coworkers for biological samples should be also mentioned. The work of Ziegel and Ashby is very important but by far not the only nor the first one. The important AI-based algorithms of the here-presented paper have been published previously in Small.

We sincerely appreciate the valuable feedback provided by the reviewer, and we would like to extend our apologies for inadvertently omitting some important references in our initial submission. It was not intentional, and we regret any oversight on our part.

We have taken the reviewer's comments seriously and have made the necessary revisions to address this concern. We have included the suggested references for the first papers on spiral scans or Lissajous scanning in the revised manuscript. Furthermore, we have added a reference to the implementation of spiral scan on Scanning Electrochemical Cell Microscopy, which showcases the achievement of high-speed imaging of electrochemical fluxes.

Regarding the work of Toshio Ando ("High-speed Atomic Force Microscope for Studying Biological Macromolecules in Action," ChemPhysChem, 2003), we acknowledge that it was mentioned in the initial manuscript while discussing the time limitations of SS-KPFM. However, we understand the importance of providing comprehensive references, and we have now included the reference to the development of high-speed AFM in our revised manuscript ("A high-speed atomic force microscope for studying biological macromolecules," PNAS, 2001).

Once again, we deeply appreciate the reviewer's efforts in strengthening the background literature of our work, and we are grateful for their valuable insights.

In summary, this reviewer sees the novelty of the present approach in combining existing techniques, albeit on the highest level. However, taking this point into consideration a more specialized journal would be more appropriate as the manuscript is centered around the technique and not the materials. The manuscript is otherwise excellent and merits definitely publication!

We acknowledge and appreciate again the positive assessment of the manuscript by the reviewer. We would like to reemphasize that the major novelty of this work lies not in spiral scanning or the image reconstruction techniques, but rather in the substantial advancement of imaging rate for charge dynamics when both are combined. The significant boost in imaging speed achieved, on the order of magnitude (e.g., ~100 milliseconds vs. minutes per image), compared to the gold standard methods, allows us to glean additional valuable material insights from the enhanced temporal resolution.

To address the reviewers' legitimate concerns, we have made significant efforts to incorporate additional experimental and theoretical contributions that enhance our understanding of the novel material dynamics taking place. As a reminder, our study focuses on two samples, specifically an LAO/STO planar device and TiO₂ thin films. We choose these materials for their relevance as well as to demonstrate the versatility of our new method in probing both devices and materials. The LAO/STO heterointerface is particularly intriguing due to its unique electronic properties, along with its potential applications in oxide electronics and quantum phenomena. However, it should be emphasized that further exploration is required to fully comprehend the coupling between LAO surface adsorbates, mobile ions, and the electronic properties at the buried LAO/STO interface for real-world applications. In this study, by comparing macroscale device testing (in relation to humidity) with the SS-KPFM technique, we establish a direct correlation between the charge dynamics measured by SS-KPFM and redox-mediated charge diffusion demonstrating a viable path to linking device

performance and nanoscale processes. See new Supporting information S9, where the slower characteristic time constants of the system show a strong dependency on environmental humidity:

Figure S9: a) Nyquist plot of the LAO/STO planar device for different RH. b) Characteristic times of the device derived from a.

Furthermore, we demonstrate that through local modification of polycrystalline TiO₂ films using a biased probe at different temperatures, we observe a noticeable alteration in the rate of charge diffusion. By comparing our experimental results with molecular dynamics (MD) simulations, we establish a direct connection between the charge dynamics and the transport of oxygen vacancies within the bulk of the material. Through fitting the experimentally measured relaxation as a function of temperature, we extract an activation energy of 0.18 eV. Remarkably, as we describe in the resubmitted manuscript, this value aligns with the findings reported in the literature (0.35 eV at 0K without an electric field), which is strengthened further by our simulations that show that if we take into account that the high electric field beneath the tip, we can expect an effective reduction in the diffusion barrier compared to the prediction in the absence of an electric field.

See new Figure 5 for more details:

Figure 5: Activation energy and temperature dependent oxygen vacancy diffusion. *a*) CPD maps after application of -8V during 1 minute at different times and for different temperatures. *b*) Corresponding profiles along the dashed black line over time. *c*) CPD evolution averaged over the central region (dashed squares in *a*) at 3 different temperatures. *d*) Arrhenius type fit to find the activation energy (E_A). *e*) DFT calculated external electric field-dependent formation energy of oxygen vacancies on a (110) TiO_2 rutile surface. The surface vacancy (V_{O1} , or 0.0 Å) is marked in red, nearest subsurface vacancy (V_{O2} , or 2.6 Å) in green, and deeper vacancy (V_{O3} , or 5.0 Å) in blue. The distance reported is distance from the surface.

Overall, these expanded research aspects contribute to a more comprehensive and insightful analysis of surface charge dynamics, enhancing the scientific significance and overall quality of our work and surpassing the novelty barrier that the reviewer was concerned about.

Reviewer #2:

The work entitled “High speed mapping of surface charge dynamics via Spiral Scanning Kelvin Probe Force Microscopy” by Checa et al. nicely renders the importance of resolving dynamic processes not only in terms of topographical changes, but also as a function of their electronic properties in time and length scales. If the comments above are taken into account, then I truly recommend this work for publication. It is an interesting approach to push the limits of scanning probe microscopy towards time depended material contrast observations. Other than that I really like the presented work.

Dear Reviewer,

Thank you very much for your thoughtful and encouraging review of our manuscript. We are delighted to hear that you appreciate the importance of resolving dynamic processes in terms of both topographical changes and electronic properties in time and length scales.

We are grateful for your constructive comments and will make sure to address them to improve the manuscript. Your recommendation for publication means a great deal to us, and we are pleased that you find our approach interesting and innovative in pushing the limits of scanning probe microscopy towards time-dependent material contrast observations.

Below we present a detailed point-by-point response to his/her comments.

1. **“These techniques are known to provide valuable insights into surface charges, dielectric properties, and the contact potential difference (CPD), which represents the difference in work function between the tip and the sample.” After this section, I recommend to include two sentences, which are focused on the achievements of KPFM at the atomic scale. Here I would recommend the following citations:**

“Gross, L.; Mohn, F.; Liljeroth, P.; Repp, J.; Giessibl, F. J. & Meyer, G. Measuring the Charge State of an Adatom with Noncontact Atomic Force Microscopy *Science*, 2009, 324, 1428-1431”.

“Mohn, F.; Gross, L.; Moll, N. & Meyer, G. Imaging the charge distribution within a single molecule *Nature Nanotechnology*, Nature Publishing Group, 2012, 7, 227-231”, and

“Fatayer, S.; Albrecht, F.; Zhang, Y.; Urbonas, D.; Pena, D.; Moll, N. & Gross, L. Molecular structure elucidation with charge-state control *Science*, *Science*, American Association for the Advancement of Science, 2019, 365, 142-145”.

This would allow to reader to learn about the capabilities of modern KPFM.

We appreciate the important reviewer comment and we have added the suggested references focused on the achievements of KPFM at the atomic scale, which improve the readability and show the spatial resolution limits of KPFM technique complementing the previous references talking about the temporal resolution limits.

2. **“The bandwidth capabilities of H-KPFM were previously investigated by Garret et al., who reported a remarkable improvement in imaging speed of more than 100-fold compared to**

conventional FM-KPFM.” I would prefer to have a repetition of the citation directly behind “Garret et al.”.

We appreciate the important reviewer comment and we have added the repetition of the citation directly behind the “Garret et al.” in the main text.

- 3. “From each sparsely sampled region, ...” Can the authors elaborate more on the term “sparsed sampling”. In my understanding, sparsed sampling includes the a priori knowledge of certain image properties. The spiral however, samples not the complete real space without the knowledge of the "feature" distribution etc. In my opinion, this is a reduced sampling, not a sparse sampling.**

We appreciate the important reviewer comment. Here, we use the mathematical definition of a sparse matrix. This is, we have an image (matrix) in which most measurements are zero. Such a matrix is termed as sparse matrix and hence we have sparse measurements of the image.

We have added a sentence in the revised manuscript explaining the mathematical definition of sparse matrix to avoid confusion regarding a priori knowledge of certain image properties:

“In a separate avenue of research, the concept of sparse sampling has been critical to characterization tools in diverse fields such medical imaging⁴⁷, electron microscopy⁴⁸ as well as astronomy⁴⁹. For clarity purposes, we refer to the term sparse in its mathematical definition; that is an image (matrix) in which most measurements (unscanned regions) are zero.”

- 4. “... controlled via FPGA using a custom-built LabVIEW/python interface that drives the tip ...” What is the speed limit for this software approach?**

We appreciate the important reviewer comment. Two main limitations can arise from the use of FPGA control of the AFM:

1. First, there is the limitation of data acquisition, that is how fast can the cards read an input or send out an output signal, which for our case its limited by the IO rate of the FPGA, which is 4MS/s, meaning practically that the FPGA control is sufficient to work in the 1 kHz-1MHz range of operation of most electrical AFM modes.
2. Second, there is a limitation of control time, that is the communication time between the different software elements controlling the FPGA (in our case, mainly LabVIEW). In this case, it is difficult to give an exact number, because it depends on many different parameters, and they can all be optimized to boost speed it if necessary. Some of those parameters can be, the memory of the computer used (64MB in our case), the external control of the LabVIEW software (which we do by python but could be optimized if done directly in C++) etc. However, we have not observed any limitations from that side so far in our implementation.

Overall, we do not think the software places any relevant temporal bottleneck for the approach developed here.

- 5. “... The image was acquired using a tip trajectory of a spiral with 24 internal cycles and an imaging speed of 4 seconds per frame. ...” Have more scan geometries been tested, e.g. smaller pitch sizes / more cycles per frame?**

We appreciate the important reviewer comment. Yes, different scan geometries have been tested and are shown in the different main manuscript figures and supplementary material. For instance, in Figure 2a, 2c, and 2e different number of cycles per frame are shown, 24, 12 and 6, respectively. Also, different scan sizes are shown in Figure 2a-f ($14\mu\text{m}$), Figure 2j-m ($2.2\mu\text{m}$), Figure 3 ($16\mu\text{m}$), Figure 4 (400nm) and the new Figure 5 ($1\mu\text{m}$). For this work, we have focused on spiral trajectories, but one of the advantages of such implementation using FPGA driven control of the XY AFM piezo controllers is that it allows for tunability of scan path to any desired waveform.

To show such capabilities, we have captured 2 additional images over the same region, in the WS_2 sample from Figure 1, with similar level of sparsity, but one with spiral scan trajectory and the other one following a Lissajous (LJ) curve:

We can observe how for this concrete structure and sparsity, the spiral scan trajectory seems to do a better job at reconstructing the CPD map than the LJ trajectory, the answer as to why this is the case it is not straight forward as many things can be causing this. On the one hand, different sample geometries or structures, would result into different optimal tip trajectories for an optimized fast information capture. For instance, samples containing most relevant information close to corners of the scanned area, would be more benefitted by the Lissajous trajectories, whereas samples where information is confined in the center would benefit from spiral scans. Additionally, the hyperparameters of the inpainting algorithms (GP in this case), might have to be tuned again independently for each different scan path followed during the data acquisition.

We have added the new data with the Lissajous scan in the Supplementary information S3.

- The authors should include some more of their experimental parameters and conditions. I miss the resonance frequencies of the cantilevers, the oscillation amplitude in nm used for imaging. Also the types and manufacturers of the used cantilevers does not seem to be given.**

We are grateful for the reviewer's valuable comment, and we have taken it into consideration during the revision process. As per the suggestion, we have included a dedicated section in the Materials and Methods called "Experimental AFM." In this section, we provide a comprehensive description of all the essential experimental parameters utilized for imaging, as well as the environmental conditions in which the experiments were conducted. This addition aims to provide clarity and ensure reproducibility of our experimental procedures. Thank you for highlighting this important aspect, and we appreciate your contribution in enhancing the manuscript.

Reviewer #3:

In this contribution, a new mode of Kelvin probe force microscopy is introduced. Principles of the method and differences to existing modes are explained well and advances beyond the state-of-the-art are clearly shown. These advances are highly relevant to the field as shown in some characteristic examples from research on materials for electronic or energy applications. A remarkable impact of this work can be expected, and I support a publication in Nature Communications.

Dear Reviewer,

We sincerely appreciate your time and effort in reviewing our manuscript, and we are thrilled to receive such positive feedback on our contribution to the field. It is gratifying to know that you found our explanation of the new mode of Kelvin probe force microscopy, its principles, and differences to existing modes to be clear and well-presented.

Your recognition of the advances we demonstrated beyond the state-of-the-art and the relevance of our work to materials for electronic or energy applications is highly encouraging. We are honored by your belief in the potential for a remarkable impact and your support for publication in Nature Communications.

Some aspects listed in the comments below, however, may deserve the authors' attention to sharpen the focus of the work and increase readability for a widely interested readership. Some of the mechanistic explanations appear quite speculative and handwaving. Such details may either be omitted or, in case they are regarded relevant, be explained in a more explicit way.

We acknowledge and appreciate the feedback by the reviewer and have worked diligently to improve the mechanistic applications including significant changes to the manuscript as well as the incorporation of new data. Below we present a detailed point-by-point response to his/her comments.

- 1. The graphical abstract Figure is very busy, although it very nicely summarizes the work. It would help to use different color codes for CPD and time. Further, the meaning of the vertical axis stays unclear. I would probably omit the whole plot of "time dynamics" (anyway "space" would be the appropriate axis assignment) and the two V_0 and τ maps to focus on the central message: spiral scanning plus GP in-painting leads to fast mapping of CPD.**

We appreciate the important reviewer comment. We have considered that with the addition of the new data regarding the SS-KPFM temperature dependent experiments and subsequent activation energy obtained, the graphical abstract should be more oriented to the physicochemical material parameters obtained with the technique than the technique details itself. Therefore, we have so far changed it to the following:

However, after acceptance of the manuscript for publication we expect our graphical team to make a more realistic rendered version of it.

2. It should be considered throughout the text that the term “dynamics” by definition contains a temporal change. Therefore, an individual CPD map cannot contain any dynamics, just a time-series of such maps does (see, e.g., last sentence of abstract).

We appreciate the important reviewer comment. Therefore, in the revised manuscript we have removed the term “charge dynamics” in situations where we refer to a single CPD map, like the last abstract sentence of the first submitted version.

3. It would be useful to indicate that the minimum time to measure a KPFM image of 16s mentioned on the top of page 2 given by the limitation of the topography feedback loop would be considerably shorter for planar samples. Reference to this thought would also be relevant on the bottom of page 4, where the analysis of planar samples is discussed.

We thank the reviewer to bring up such an important question. In Garret et al.*, the limitation of 16s per image is performed on a few layers graphene (FLG) layer of height 2nm. Our discussion of the scanning speed and sparsity in Figure 2 is performed on a WS₂ layer on Si sample with a surface roughness of 2nm, and on a PZT thin film of surface roughness 3nm. Therefore, our sample has similar sized structures as the one in Garret et al.* , enabling a relevant basis for comparison.

Figure S6: a) Topography of WS₂ flake of Figure 2 of the main manuscript. b) Topography of PZT of Figure 2 of the main manuscript.

We have added an additional Figure in Supplementary information S6 showing the topography images simultaneously acquired of the WS₂ flake and the PZT film showing it and a sentence in the main text for clarification.

* Garrett, J. L. & Munday, J. N. Fast, high-resolution surface potential measurements in air with heterodyne Kelvin probe force microscopy. *Nanotechnology* 27, 245705 (2016).

4. The time scale should be given in a unified way. Either the number of frames per second, or the time needed for a measurement or the time resolution of a physical quantity (e.g. CPD) should be used. This will help to compare different statements in the manuscript and to follow comparison to references.

We appreciate the important reviewer comment. We have unified both in the text and the figures, every time scale to frames per second (fps), so that they can all be compared properly.

5. In the direct comparison of Figure 1, the raster method seems to provide the superior spatial resolution. It should be indicated in the Figure or the legend, that the temporal resolution is far better in the spiral scan approach. Also in Figure 2, the time needed for the raster image would be nice to see. Perhaps a) the time needed to measure the same area as that covered by the spiral or b) the time needed to measure ~15 "checkerboard fields" would provide a valid comparison.

We appreciate the important reviewer comment. Therefore, in the caption of the figures of the revised manuscript we have specified the temporal resolution of the spiral scan approach in Figure 1 and the time needed to acquire the raster image in Figure 2.

However, we think that the time needed to acquire the raster-scan images displayed in Figure 1 and Figure 2 should not be compared with the times per frame of the SS-KPFM, as such raster-scan images were not pushed to its temporal limit, but were taken to have a measurement of the “ground truth” to which compare the accuracy of the CPD. That is the reason why we always compare our time resolution to the one of the fastest (to date) published KPFM image in *. We have added a sentence to clarify this subject.

* Garrett, J. L. & Munday, J. N. Fast, high-resolution surface potential measurements in air with heterodyne Kelvin probe force microscopy. *Nanotechnology* 27, 245705 (2016).

6. The environment for the measurements (air, inert gas, vacuum?) needs clear specification. Otherwise, explanations about water layers, ionic movements etc. lack relevance.

We appreciate the important reviewer comment and we have added a section in the Materials and Methods called *Experimental AFM* where we specify all those important experimental parameters used for imaging. All the AFM experiments were performed in air at room temperature (25°C) except for the temperature-dependent measurements of Figure 5, where a temperature control sample holder was used. For the temperature-controlled experiments, the system was left for 20 minutes every time the temperature was changed to let the system thermalize and avoid drift. In addition, in the revised manuscript bulk impedance measurements were added as a function of humidity (see new Supplementary Information S9).

7. Assignments of figure parts and legend are not consistent in Figure 3. Further, in the

experimental part Ni is mentioned, Au is indicated in the Figure. It would help if the position 80 would also be shown in parts d) and f), or, even better, if the pixel chosen for e) and g) would be positioned on the line for which data are plotted in d) and f). The text “j) and k) ... point by point.” needs to be omitted.

We thank the reviewer to notice our mistake. He/she is completely right, the *Au* labels in Figure 3 are wrong and have been changed for the correct *Ni* ones. Also, the *Source* and *Drain* Labels have been added for clarification:

Figure 3: SS-KPFM on LAO/STO lateral device. *a)* Schematics of the experimental setup. *b)* and *c)* CPD maps of LAO/STO – and part of the grounded electrode - during negative and positive biasing of the opposite electrode ($V_{SD}=-6V$ for *b* and $V_{SD}=+6V$ for *c*) respectively. *d)* CPD profiles over time during 78 seconds for $V_{SD}=-6V$ and $V_{SD}=+6V$, extracted from sections of all the CPD frames along the grey dashed lines. *e)* and *g)* are CPD evolution over time for a single pixel (marked in purple: number 80,64). Green dashed line indicates the application of the V_{SD} bias pulse. Red dashed line shows the exponential decay fit. Black circle indicates V_0 . *h)* and *i)* are the V_0 maps and *j)* and *k)* are the τ maps extracted from the fits performed point by point. *j)* and *k)* are the corresponding τ maps.

We also want to thank the reviewer for the nice suggestion in the rearrangement of Figure 3. In the revised manuscript, Pixel (80,64) is shown instead of (80,80), which coincides with that of the gray dashed lines shown in d) and f). Additionally, a purple vertical line is shown in d) and f),

which indicates the position of the (80,64) pixel shown in e and g. The “point-by-point” was removed at the end of the Figure 3 legend.

8. On page 9, it would help to clarify which electrode is referred to as source/drain.

We appreciate the important reviewer comment, we have explicitly said it in the text which electrode is biased and which one is grounded and have also added the source and drain labels in Figure 3, clarifying that the grounded electrode is the one we refer to as the drain and the biased one is the one, we refer to as the source.

9. The explanations provided from the bottom of page 10 either should be omitted (if the focus is just placed on the new KPFM method) or need more specific reference to prior literature on LAO (if the reader should have a chance to follow the authors’ argument).

We thank the reviewer for the comment. The focus of the resubmitted manuscript is placed both on the implementation of a new KPFM method, but also in the type of material information we can extract from it. See answer to the next two questions for more details on the author’s physicochemical interpretation of the results.

As a reminder, our study focuses on two samples, specifically an LAO/STO planar device and TiO₂ thin films. We choose these materials for their relevance as well as to demonstrate the versatility of our new method in probing both devices and materials. The LAO/STO heterointerface is particularly intriguing due to its unique electronic properties, along with its potential applications in oxide electronics and quantum phenomena. In our resubmission, we have added macroscale device testing as a function of humidity that can be coupled to the results obtained at the nanoscale with the SS-KPFM, to establish a direct correlation between the charge dynamics measured by SS-KPFM and a surface water sensitive electrochemically mediated charge diffusion process demonstrating a viable path to linking device performance and nanoscale processes. However, it should be emphasized that further exploration is required to fully comprehend the chemical nature of such water driven surface electrochemistry if specific reactions want to be studied. We speculate those can be:

Surface water interact with LAO surface oxygen ions: surface water chemisorbed fill oxygen vacancies present in the LAO.

...

For further details we refer to the following references:

Dling, J. et al. Spatially resolved probing of electrochemical reactions via energy discovery platforms *Nanoletters* (2015).

Strelkov, E. et al. Direct Probing of charge injection and Polarization-Controlled Ionic Mobility on Ferroelectric LiNbO₃ Surfaces. *Advanced Materials* (2013).

Furthermore, we demonstrate that through local modification of polycrystalline TiO₂ films using a biased probe at different temperatures, we observe a noticeable alteration in the rate of charge diffusion. By comparing our experimental results with molecular dynamics (MD) simulations, we establish a direct connection between the charge dynamics and the transport of oxygen vacancies within the bulk of the material. Through fitting the experimentally measured relaxation as a function of temperature, we extract an activation energy of 0.18 eV. Remarkably, as we describe in the resubmitted manuscript, this value aligns with the findings reported in the literature (0.35 eV without an electric field), which is strengthened further by our simulations that show that if we take into account that the high electric field beneath the tip, we can expect an

effective reduction in the diffusion barrier compared to the prediction in the absence of an electric field.

10. The meaning of the results discussed on page 11 strongly depend on the medium the films are in contact with (see comment 6). For electrochemical impedance measurements, a liquid contact is typically used. This is not mentioned in the text. Diffusion of ions is also compared to diffusion coefficients of ions in (diluted) aqueous solutions. Is there a water film present for which a similar role can be assumed? Otherwise, ions bound to the oxide surface would have to be discussed. Was there any influence of humidity studied? Is there a distinguished meaning of positive or negative polarization against an external reference (electrode?) or is it just meaningful for one electrode relative to the other? Similar: any meaning of grounding? Any chemical reactions possible under the given experimental conditions? A rather complex discussion is opened but not concluded to satisfaction. Again, it could either be omitted for the sake of focussing on the method and neglecting the aspects of materials science or the details needed to follow the discussion should be added. At present the role of different charge carriers (electrons, anions, cations) remains unclear and the readership has no chance to develop a clear understanding which “faradaic processes” or “surface electrochemical activity” is possible under the given circumstances. At present, the arguments stay handwaving. If the authors have a clear working hypothesis backed up by the literature they are welcome to explicitly present it. Otherwise, it may be better to leave such analysis to detailed future measurements. Such explanation at least is not mandatory at present to show the benefits of spiral scanning in KPFM. If understood correctly, the voltage drop is maximum “in front of” or “close to” the more positive electrode, not “in” it.

We appreciate the important reviewer comment. We agree that the previous submission of the work was lacking in solid interpretation of the material dynamics. Therefore, we performed additional measurements to enable us to give a more solid interpretation of the phenomena that can be visualized with our approach in each one of the presented applications. For the case of the LAO/STO results in Figure 3, those are slow ($\tau \approx$ seconds) surface electrochemical reactions that can be mapped spatially at the nanoscale.

Find the response to each question raised by the reviewer analyzed separately:

- i) The bulk measurements shown in supplementary information are bulk impedance measurements, not bulk electrochemical impedance measurements, like we wrote in the previous version of the manuscript, so no liquid contact is used at all. There is not a specific meaning of positive or negative polarization against an external reference electrode, the laterally applied biases are only relative to the source and drain (I guess this question was related to the confusion generated by the electrochemical impedance but does not have any further relevance now that this mistake is corrected). We thank the reviewer again and we apology for the confusion generated in the previous manuscript version. We have corrected the mistake in the revised one eliminating the word “electrochemical”.
- ii) We have added bulk impedance measurements as a function of humidity for the LAO/STO devices which nicely show that the observed the “fast” characteristic times get retarded with increasing relative humidity, indicating that the presence of more water on the surface facilitates faster migration of H^+ and OH^- groups. Moreover, the “slow” characteristic times only appear at high RH levels and seem to disappear under low RH

conditions, emphasizing the role of water in influencing the system's slower behavior (which is the one mapped by the SS-KPFM), as higher RD makes it easier to trigger electrochemical reactions or faradaic processes. See new S9:

To avoid complication in the discussion, we have removed the comparison with the diffusion coefficient of ions in bulk electronic solutions.

- iii) The SS-KPFM measurements were performed in ambient air at room temperature and without humidity controlled. However, at ORNL (located in east Tennessee in the USA) humidity oscillates between 55% and 85% all year round, therefore we expect a thin water film to be present on our devices⁴.
- iv) There are a variety of chemical reactions possible under the given conditions of humidity, temperature and applied bias. In the manuscript we reference to^{5,6} where they are numbered and studied in more detail with similar techniques. However, some possible chemical reactions related to the presence of a water film possible are:

Surface water interact with LAO surface oxygen ions: surface water chemisorbed fill oxygen vacancies present in the LAO.

...

KPFM it is probably not the right technique to be able to discriminate what happens as it is “chemically blind”; however, it can be the right technique to locally visualize their presence, and as we have showed, when coupled to bulk impedance measurements can allow the study of threshold biases, and diffusion coefficients or activation energies, among others. Additionally, in the Supplementary Information S7, the bulk impedance measurements show the presence of a certain activation barrier needed to trigger such “slow” electrochemically mediated response, which also points towards the surface water electrochemistry.

- v) “close to” it is added when writing about the voltage drop location, instead of “in”.

¹Collins, L. et al. Visualizing Charge Transport and Nanoscale Electrochemistry by Hyperspectral Kelvin Probe Force Microscopy. *Appl. Mat. & Interf.* (2020).

²Collins, L. et al. Multifrequency spectrum analysis using fully digital G Mode-Kelvin probe force microscopy. *Nanotechnology* (2016).

³Murawski, J. et al. Pump-probe Kelvin-probe force microscopy: Principle of operation and resolution limits. *Journal of Applied Physics* (2015).

⁴Ewing, George E. "Ambient thin film water on insulator surfaces." *Chemical reviews* 106.4 (2006): 1511-1526.

⁵Dling, J. et al. Spatially resolved probing of electrochemical reactions via energy discovery platforms *Nanoletters* (2015).

⁶Strelkov, E. et al. Direct Probing of charge injection and Polarization-Controlled Ionic Mobility on Ferroelectric LiNbO₃ Surfaces. *Advanced Materials* (2013).

11. Explanation of the TiO₂-results lack consistency. O₂ is not an ion. Negative species must be thought of to be driven into the lattice (e.g. O with the charge of -2?) at negative polarization or positive species pulled out (such as vacancies of negatively charged O-ions). To consider such vacancies being pulled out at positive bias appears counter-intuitive. Further, which evidence do the authors have for moving O-ions and vacancies as opposed to electrons, H⁺ or OH⁻? In these experiments, it is not explained why a voltage pulse at 8V can lead to a higher CPD than a pulse at 10 V. A higher CPD indicates a higher concentration of charges for +8V. It is unclear, however, why such higher concentration would not diffuse across the grain boundary, but the smaller concentration at +10 V does. Rather, a barrier could be overcome at 10V and the process was too fast to resolve it in the present experiments. It cannot be stated that the initial surface charge state is recovered within the measurement window since significantly different patterns are left after 140 s at either +10V or -10V. Subtraction of the line measured at 140s from all lines (background correction) would probably also show this very clearly.

We appreciate the important reviewer comment. To achieve a better understanding of the mechanisms involved in the experiments performed with the TiO₂, we have performed new SS-KPFM experiments as a function of temperature that enabled us to obtain activation energy value for the nanoscale diffusion process that is being imaged (obtaining a 0.18eV value, in good agreement but slightly below previously DFT calculated oxygen vacancy diffusion in TiO₂, ≈0.35eV). We have also included additional molecular dynamics simulations that show that if we consider the high electric field beneath the tip drives the oxygen ions deeper into the material, we can expect an effective reduction in the diffusion barrier compared to the prediction in the absence of an electric field.

Find the response to each question raised by the reviewer analyzed separately:

- i) Due to the addition of new experiments the old Figure 4 has been moved to the Supplementary Information and it is now Figure S10. The authors interpretation of the higher CPD for the +8V injection as compared to the +10V case is the following. Charge migration and diffusion within a single TiO₂ grain has a lower energy barrier than between grains. Therefore, when the field is applied, charges will be redistributed in the same grain (+8V case), unless a certain energy barrier is exceeded (+10V case). If that energy barrier is exceeded (+10V case) and charges can migrate/diffuse to a neighboring grain, by charge conservation, the charge present in a single grain will now have to be divided into 2 grains, what will result in a lower local charge concentration, that will lead to a lower CPD.

Additional experimental data to prove our explanation has been added in the Supplementary Information of the revised manuscript. We considered 2 neighboring grains and injected charge at each grain separately (a, d,c,f) and at the grain boundary (b,e) achieving simultaneous charge injection in both grains. See new Supplementary Information S12:

Figure S12: Topography images corresponding to the charge injection at the left grain (a), at the grain boundary, touching both grains (b) and at the right grain (c). Corresponding CPD maps with charge injection at the left grain (d), at the grain boundary, touching both grains (e) and at the right grain (f).

We can observe that for the injection/redistribution in a single grain (d and f), the CPD gets higher than when the charge injection is performed simultaneously at both grains, as charge must divide between both grains resulting into a lower CPD. We have added the additional experimental work performed in a new Supplementary Information section, S12.

- ii) As the reviewer stated, for the negative voltage applied (Figure S10h, i and j), the initial charge state is not fully recovered within the measurement window (the first 140s after the charge is injected). However, with the new temperature-dependent measurements added in the resubmission the old Figure 4 is moved to the Supplementary material (Figure S10) so such text is not in the manuscript anymore.
- iii) The temperature dependent measurements can be seen in the new Figure 5:

Figure 5: Activation energy and temperature dependent oxygen vacancy diffusion. *a)* CPD maps after application of -8V during 1 minute at different times and for different temperatures. *b)* Corresponding profiles along the dashed black line over time. *c)* CPD evolution averaged over the central region (dashed squares in *a*) at 3 different temperatures. *d)* Arrhenius type fit to find the activation energy (E_A). *e)* DFT calculated external electric field-dependent formation energy of oxygen vacancies on a (110) TiO_2 rutile surface. The surface vacancy (V_{O1} , or 0.0 \AA) is marked in red, nearest subsurface vacancy (V_{O2} , or 2.6 \AA) in green, and deeper vacancy (V_{O3} , or 5.0 \AA) in blue. The distance reported is distance from the surface.

The impact of temperature on the charge diffusion process is evident, with subsequent charge relaxation towards the pristine state occurring faster at higher temperatures and slower at lower temperatures. The averaged CPD evolution (Figure 5c) over the central region (colored dashed squares in Figure 5a) represent the mean response of the center of the grain for each different temperature, where the temperature evolution can be clearly observed. Furthermore, we applied an Arrhenius-type law to fit the characteristic diffusion time derived from the SS-KPFM results (Figure 5d). This fitting procedure yielded a value for the activation energy (E_A) of the charge diffusion process, which was found to be 0.18eV . Our measured activation energy is similar to the diffusion barrier predicted by first principles (0.35 eV)¹. The somewhat lower value

measured by our experiments may be explained by the temperature mismatch between the calculations (performed at 0K)¹ and the experiment, and by the presence of an external electric field, which would lead to surface charging and altered lattice strain caused by local geometry reorganization. We therefore use DFT calculations in the Vienna Ab initio Simulation Package (VASP) to explore the possibility of external electric fields reducing the V_O diffusion barrier in our experiments. The formation energy for an oxygen vacancy on a (110) surface (V_{O1}, or 0 Å below the surface) was calculated at the GGA+U-level and compared with the energies of a subsurface vacancy 2 layers (~2.6 Å) below the surface (V_{O2}), and a subsurface vacancy 4 layers (~5.0 Å) below the surface (V_{O3}, Fig. 5e), finding that the presence of the electric field induces a reduction in the diffusion barrier of the oxygen vacancies.

¹Wu, Z. et al. Oxygen Vacancy Diffusion in Rutile TiO₂. *The Journal of Physical Chemistry Letters*. (2023).

12. Structural and elemental characterization of the sample materials are missing. If the authors have prepared the same materials previously under identical conditions (as it appears from the text) reference to such earlier work would suffice.

We appreciate the important reviewer comment, and we have included in the revised manuscript either additional experimental material or additional references to previously published material using identical conditions of the sample materials used in this manuscript. Those are:

1. For the WS₂ flakes in Silicon from Figure 1 and 2:

Optical, Photoluminescence (PL) and additional AFM characterization → G. Hwi An et al. Growth Mechanism of alternating defect domains in Hexagonal WS₂ via inhomogeneous W-Precursor Accumulation. *Small* (2020).

2. For the PZT thin films of Figure 2:

XRD reciprocal space mapping (RSM) and additional PFM characterization → J.Y. Wang et al. Morphology-dependent photo-induced polarization recovery in ferroelectric thin films. *Appl. Phys. Lett.* (2017).

3. For the LAO/STO lateral devices of Figure 3:

X-ray diffraction analysis (XRD) and reciprocal lattice mapping (RLM) characterization of the LAO/STO structure → W. Wei et al. Strain relaxation analysis of LAO₃/SrTiO₃ heterostructure using reciprocal lattice mapping. *Appl. Phys. Lett.* (2012).

High-Angular Annular Dark-Field (HAADF), Energy Electron Loss Spectroscopy (EELS) and Medium Energy Ion Scattering (MEIS) characterization of the LAO/STO structure → H. Zaid et al. Atomic-resolved depth profile of strain and cation intermixing around LaAlO₃/SrTiO₃ interfaces. *Scientific Reports* (2016).

Electrical measurements of the LAO/STO structure → H. Zaid et al. Role of the different defects, their population and distribution in the LAO₃/SrTiO₃ heterostructure's behavior. *Journal of Applied Physics* (2018).

4. For the TiO₂ thin films of Figure 4:

SEM and XRD patterns of the sample have been measured and added in the Supplementary information S13:

Figure S13: (a, b) SEM images of the polycrystalline TiO_2 on the Pt/Ti/SiO₂/Si substrate: (a) cross-section and (b) surface. (c) XRD patterns of the polycrystalline TiO_2 . The XRD pattern of the polycrystalline TiO_2 in Figure xc clearly shows (211), (220), and (002) peaks of rutile.

Reviewer #4:

The manuscript “High speed mapping of surface charge dynamics via Spiral Scanning Kelvin Probe Force Microscopy” by M. Checa et al. shows the capabilities of spiral SKPFM for high-speed acquisition surface electrical properties in nanoscale materials. The power of the technique is demonstrated on two highly interesting material systems resulting in interesting findings on the relaxation dynamics. Overall, the results are presented in an excellent way. The existing literature is reviewed well and the results are set into context. The article is well written and is certainly suitable for Nature Coms, however, a few aspects could be still improved:

Dear Reviewer,

We are grateful for your thorough review and positive feedback on our manuscript. It is encouraging to know that you found the presentation of our results excellent, and we appreciate your recognition of the power of SS-KPFM.

Additionally, we sincerely thank you for pointing out aspects that could be improved in our manuscript. Your suggestions will undoubtedly help us enhance the quality of our work, and we are eager to address those points in the revised version of our manuscript. We believe that incorporating your insights will not only strengthen our submission but also contribute to its suitability for publication in Nature Communications.

Below we present a detailed point-by-point response to his/her comments.

- 1. While the authors show clearly that the performance of the technique could be significantly improved with respect to the previous reports by Garret et al. by introducing the spiral scanning (the supporting info shows images and performance data acquired in raster scanning and spiral scanning), the discussion in the last paragraph of the spiral scanning section on page 8 the technical factors that limit the speed of the technique are only discussed superficially. It would be nice to have here or in the supporting info a discussion on what are the actual technical limitations. Is it the FPGA card you used? Which are the limiting time constants here? Is it the output bandwidth of the lock in amplifier, or the feedback itself? Can you give experimental proof of this? Which would be the route for improving the speed further? I understand that it is not the aim to make a technical article out of this, but it is important to give a perspective for further technical development.**

We appreciate the important reviewer comment, and we have expanded the discussion at the end of the spiral scanning section in page 8, where we expose the technical factors limiting technique speed and identify the possible routes for improving the speed further.

The technical limitations in terms of achieving the maximum scanning speed following the presented methodology, will depend on many different factors:

- i) Speed at which the PID loop of the H-KPFM can accurately measure the CPD, which imposes a limitation to the minimum time per pixel needed to avoid pixel averaging. The fastest time constant (or output bandwidth) of the lockin amplifier used for the CPD compensation feedback will be the constrain in this case, which ultimately will be limited by the feedback gains, which will determine the PID loop bandwidth.
- ii) Speed at which the topography feedback loop can measure sample's topography before breaking down. Topography feedback gains must be tuned accordingly. An estimation of such limitation can be extracted by calculating lateral tip speed at which topography feedback breaks down in ¹,

which is $32\mu\text{m}\cdot\text{s}^{-1}$. However, such limitation will vary depending on the specific topographic features of each sample. and our method circumvents it by sparsely scanning instead of scanning fast.

- iii) FPGA's IO rate for processing the data, as exposed before the actual setup enables 4MS/s, which does not imply any limitation.
- iv) XY piezo scanner speed to accurately perform the scan (especially relevant in XY open loop configuration to avoid image distortion). We are currently working on a closed-loop version implementation which will read the sensor data and correct for piezo distortions on-the-fly. However for the scanning speeds and scanning sizes used in this manuscript we have not observed major distortions in our images.
- v) Accuracy of image reconstruction algorithms that enable the recovery of the ground truth using sparser scans.

Therefore, the route for improving the speed even further would be addressing the previously mentioned issues by:

- i) Using higher frequency tips, which would help increase the time limitations on the PID loops tracking both CPD and topography. High speed AFM has already shown that topographic video-rate imaging can reach >10 fps when using high resonance ($\approx\text{MHz}$) tips and a specific scan head design. However, the increase in the frequency usually comes with an increase in cantilever spring constant, which can become hurdle when trying to measure the minute electrostatic capacitive forces of KPFM.
- ii) Developing a similar strategy for open-loop or dual-harmonic KPFM², where CPD is not physically compensated in-situ, but determined by postprocessing, so no CPD feedback loop is needed.
- iii) Improving FPGA electronics to reach higher sampling rate.
- iv) Developing more accurate image reconstruction schemes, that would allow increasing sparsity, therefore reducing imaging time. The use of deep learning techniques containing more specific priors would help like it is shown in³.

¹ Garrett, J. L. & Munday, J. N. Fast, high-resolution surface potential measurements in air with heterodyne Kelvin probe force microscopy. *Nanotechnology* **27**, 245705 (2016).

² Collins, L. Dual Harmonic Kelvin probe force microscopy at the graphene liquid interface (2014).

³ Xie, Y et al. A review of deep learning methods for compressed sensing image reconstruction and its medical applications. *Electronics* (2022).

2. Concerning the section on the LAO/STO lateral capacitor: it is not fully clear if you measure the fast (40 and 80 ms) time constants in the diffusion processes. Please clarify this. Please also discuss a bit more in detail the comparison with impedance spectroscopy data. There is certainly a difference between macro and nanoscale measurements.

We appreciate the important reviewer comment. Our explanation is the following.

- i) In this work, we acknowledge that we cannot directly measure the fast time constants of 40ms and 80ms associated with fast migration processes of protons and hydroxyls in the surface water layer. This limitation arises due to our imaging frame rate of 1fps in the LAO/STO measurements. To resolve such rapid processes, spectroscopic time-resolved (tr-KPFM) measurements, similar to those conducted in our previous studies¹, or techniques like G-modem KPFM² or pump-probe KPFM³, would be required. Therefore, it is important to note that SS-KPFM is primarily sensitive to processes which characteristic times $\tau >$ imaging frame rate.

However, this regime also involves a rich landscape of slow ionic and electrochemical processes which are not accessible by these previously mentioned spectroscopic tr-KPFM approaches (see the interpretation of such slower time dynamics in the following discussion point).

- ii) Regarding the interpretation of the impedance spectroscopy data presented in the supplementary information and its comparison with the SS-KPFM data in Figure 3: As mentioned earlier, SS-KPFM is not sensitive to very fast processes with time constants above 1 second. Therefore, the fast processes observed in the impedance response, representing the rapid global responses of the device, will not be captured locally by SS-KPFM. However, processes with time constants above 1 second (such as the 9.52s process shown in Figure S9d) can be spatially resolved at the nanoscale using SS-KPFM (as seen in Figure 3j and Figure 3k, τ maps). The 9.52s time constant represents an average of all the processes occurring locally within the device, which is in good agreement with the characteristic time maps presented in Figure 3j and Figure 3k.

Additionally, as already stated in previous reviewers' answers, for the resubmission we have added bulk impedance measurements as a function of humidity for the LAO/STO devices which nicely show that the observed the "fast" characteristic times are retarded with increasing relative humidity, indicating that the presence of more water on the surface facilitates faster migration of H^+ and OH^- groups. Moreover, the "slow" characteristic times are correlated with high RH levels, emphasizing the role of surface water in influencing the system's slower behavior (which is the one mapped by the SS-KPFM), see new S9:

¹Collins, L. et al. Visualizing Charge Transport and Nanoscale Electrochemistry by Hyperspectral Kelvin Probe Force Microscopy. *Appl. Mat. & Interf.* (2020).

²Collins, L. et al. Multifrequency spectrum analysis using fully digital G Mode-Kelvin probe force microscopy. *Nanotechnology* (2016).

³Murawski, J. et al. Pump-probe Kelvin-probe force microscopy: Principle of operation and resolution limits. *Journal of Applied Physics* (2015).

Our interpretations connecting the macroscopic bulk impedance measurements with the nanoscopic SS-KPFM measurements have been in the resubmitted version of the manuscript together with the additional experiments and simulations for the TiO_2 sample.

Reviewers' Comments:

Reviewer #1:

Remarks to the Author:

The changes the authors incorporated into the manuscript convinced the reviewer that the manuscript could be published by Nature Communications. The authors did a great job in revising the manuscript and addressing the points raised by the reviewers.

The only point still not completely clear to me is how an optimum for the AI-based reconstruction is found. If the separation between the 'scan lines' is too small one loses time resolution, if it is too large one loses lateral resolution. Is there an optimum?

Reviewer #2:

Remarks to the Author:

The Authors have responded to all of my raised comments in their revision. The work is significant to the field. It now nicely compares and references the established literature. The work supports the conclusions and claims. The data analysis appears to be solid and rigid and applies to the standards of the scanning probe microscopy field. I therefore recommend this work for publication.

Reviewer #3:

Remarks to the Author:

In their revision, the authors took great care to improve the presentation and dealt with all problems risen by the comments. Conditions are clearly described and strongly support the authors' arguments. Extensive new experiments were added to provide a safe basis for the conclusions. In particular, the temperature-dependent measurements clearly indicate the relevance of barriers for charge transport. An impressive quality is reached and I consider the paper ready for publication. My only concern left, indeed, refers to the temperature-dependent measurements and their representation in the graphical abstract. In the text, the sequence of experiments could be clearer: if understood correctly, a constant temperature is established for each injection of charges and the study of their redistribution. In the text, however, one may think that charges are injected at room temperature and the redistribution is then studied at different established temperatures (unlikely and experimentally very challenging if not impossible). In the graphical abstract, it appears even as if a temperature gradient is applied across the sample. Such misunderstanding should be avoided. In my opinion, sparse scanning and computer-aided construction of the full dataset (GP) represent the core of the method. Then, different examples are presented to make good use of this method: checkerboard PZT thin film, LAO/STO lateral contacted device, polycrystalline TiO₂. For my perception, a simplified version of Figure 1 a/b/c (no sample scheme, no raster-KPFM, no line scans) complemented by, e.g., Figure 2m, Figure 3a and Figure 4a/b would provide a perfect start into reading the paper. But I am sure the authors will find their best solution. In any case, I am looking forward to see the work published.

Reviewer #4:

Remarks to the Author:

All my previous comments have been addressed. I recommend publication.

Reviewer #1:

The changes the authors incorporated into the manuscript convinced the reviewer that the manuscript could be published by Nature Communications. The authors did a great job in revising the manuscript and addressing the points raised by the reviewers.

Dear Reviewer,

We are grateful for your thorough review and positive feedback on our manuscript. It is encouraging to see that you think we addressed all the points raised by the reviewers. Additionally, we sincerely thank you for pointing out aspects that could be improved in our manuscript during the review process. Your suggestions have undoubtedly help us enhance the quality of our work.

The only point still not completely clear to me is how an optimum for the AI-based reconstruction is found. If the separation between the 'scan lines' is too small one loses time resolution, if it is too large one loses lateral resolution. Is there an optimum?

We appreciate the interesting point raised by the reviewer. He/She is completely right, in SS-KPFM there will be always a tradeoff between time resolution and spatial resolution. That is: the sparser the measurement the faster you can acquire each frame so the more temporal resolution, but to the cost of losing spatial accuracy. Therefore, the determination of the optimum sparsity cannot be defined in general as it will depend on the minimal necessary temporal resolution to gain insight into the dynamics of the physical phenomena under study and the minimal necessary spatial resolution to resolve the sample heterogeneities.

Practically, the way we make it work is by performing one dense scan in normal raster mode before the SS-KPFM measurement to check for the spatial features present in the sample, from which the minimal spatial resolution can be inferred/estimated, which again it will be sample dependent. For the determination of the optimal temporal resolution, either previous sample knowledge, simulations or multiscale characterization using bulk techniques can give an estimate for the order of magnitude for the temporal resolution.

Reviewer #2:

The Authors have responded to all of my raised comments in their revision. The work is significant to the field. It now nicely compares and references the established literature. The work supports the conclusions and claims. The data analysis appears to be solid and rigid and applies to the standards of the scanning probe microscopy field. I therefore recommend this work for publication.

Dear Reviewer,

We are grateful for your thorough review and positive feedback on our manuscript. It is encouraging to know that you found the work significant to the field, and solid in our conclusions and claims. Additionally, we sincerely thank you for pointing out aspects that could be improved in our manuscript during the review process. Your suggestions have undoubtedly help us enhance the quality of our work.

Reviewer #3:

In their revision, the authors took great care to improve the presentation and dealt with all problems risen by the comments. Conditions are clearly described and strongly support the authors' arguments. Extensive new experiments were added to provide a safe basis for the

conclusions. In particular, the temperature-dependent measurements clearly indicate the relevance of barriers for charge transport. An impressive quality is reached and I consider the paper ready for publication.

Dear Reviewer,

We are grateful for your thorough review and positive feedback on our manuscript. It is encouraging to know that you found the work is clearly described and our claims are strongly supported by our measurements. Additionally, we sincerely thank you for pointing out aspects that could be improved in our manuscript during the review process. Your suggestions have undoubtedly help us enhance the quality of our work.

My only concern left, indeed, refers to the temperature-dependent measurements and their representation in the graphical abstract. In the text, the sequence of experiments could be clearer: if understood correctly, a constant temperature is established for each injection of charges and the study of their redistribution. In the text, however, one may think that charges are injected at room temperature and the redistribution is then studied at different established temperatures (unlikely and experimentally very challenging if not impossible). In the graphical abstract, it appears even as if a temperature gradient is applied across the sample. Such misunderstanding should be avoided. In my opinion, sparse scanning and computer-aided construction of the full dataset (GP) represent the core of the method. Then, different examples are presented to make good use of this method: checkerboard PZT thin film, LAO/STO lateral contacted device, polycrystalline TiO₂. For my perception, a simplified version of Figure 1 a/b/c (no sample scheme, no raster-KPFM, no line scans) complemented by, e.g., Figure 2m, Figure 3a and Figure 4a/b would provide a perfect start into reading the paper. But I am sure the authors will find their best solution. In any case, I am looking forward to see the work published.

We appreciate the interesting point raised by the reviewer. We agree with the reviewer that the graphical abstract submitted in the last revision could be clearer and to avoid misinterpretation of it our graphical team has worked on an updated version of it for the final submission after acceptance:

The editorial team has asked us to remove the graphical abstract from the final version of the manuscript, but this final image will be used by Nature Communications in their highlights and social media for this article, which we think summarizes in a beautiful way the type of induced charge dynamics that sparse spiral scanning can give insight into.

Reviewer #4:

All my previous comments have been addressed. I recommend publication.

Dear Reviewer,

We are grateful for your thorough review and positive feedback on our manuscript. It is encouraging to know that you found the presentation of our results excellent, and we appreciate your recognition of the power of SS-KPFM. Additionally, we sincerely thank you for pointing out aspects that could be improved in our manuscript during the review process. Your suggestions have undoubtedly help us enhance the quality of our work.